# KS-GNN: Keywords Search over Incomplete Graphs via Graph Neural Network

**Yu Hao**
University of New South Wales
NSW, Australia
yu.hao@unsw.edu.au

**Xin Cao**[*]
University of New South Wales
NSW, Australia
xin.cao@unsw.edu.au

**Yufan Sheng**
University of New South Wales
NSW, Australia
yufan.sheng@unsw.edu.au

**Yixiang Fang**
Chinese University of Hong
Kong, Shenzhen, China
fangyixiang@cuhk.edu.cn

**Wei Wang**
The Hong Kong University of Science
and Technology, Guangzhou, China
weiwcs@ust.hk

## Abstract

Keyword search is a fundamental task to retrieve information that is the most relevant to the query keywords. Keyword search over graphs aims to find subtrees or subgraphs containing all query keywords ranked according to some criteria. Existing studies all assume that the graphs have complete information. However, real-world graphs may contain some missing information (such as edges or keywords), thus making the problem much more challenging. To solve the problem of keyword search over incomplete graphs, we propose a novel model named KS-GNN based on the graph neural network and the auto-encoder. By considering the latent relationships and the frequency of different keywords, the proposed KS-GNN aims to alleviate the effect of missing information and is able to learn low-dimensional representative node embeddings that preserve both graph structure and keyword features. Our model can effectively answer keyword search queries with linear time complexity over incomplete graphs. The experiments on four real-world datasets show that our model consistently achieves better performance than state-of-the-art baseline methods in graphs having missing information.

## 1   Introduction

Keyword search is an important research topic which allows users to provide query keywords and returns the most relevant results. The keyword search over graph data [1] usually retrieves top-$k$ subtrees or subgraphs which contain all the query keywords ranked according to some criteria. For example, He et al. [2] propose a general scoring function considering both graph structure and content, and they aim to find top-$k$ nodes where each node can reach all query keywords, and the sum of its shortest path distances to these keywords is as small as possible. This ranking method is commonly used in later graph keyword search works [3, 4].

---

[*]Corresponding author.

35th Conference on Neural Information Processing Systems (NeurIPS 2021).

All existing studies assume that the graph data is complete and has no missing information. However, real-world graphs usually have some missing edges [5] and missing attributes on some nodes [6]. This renders previous graph keyword search methods unable to find exact answers when dealing with such incomplete graphs. Figure 1 shows an example keyword search query over graphs. Given $q=\{c, e, f\}$, on the left graph $G$ with no missing information, the best node is $v_4$, since it contains keywords $c$ and $f$, it can reach $v_5$ containing $e$, and its sum of the shortest path distances to all query keywords is the smallest which is 1 (the shortest distances of $v_4$

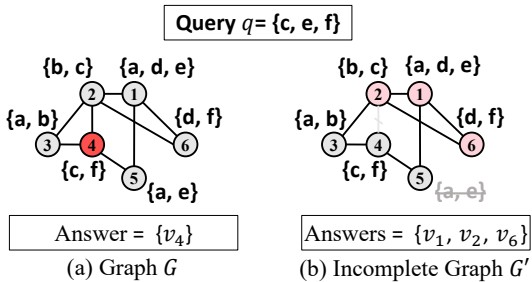

Figure 1: Example of keyword search on incomplete graphs

to $c$, $e$, and $f$ are 0, 1, and 0). However, on the right graph $G'$ which has a missing edge and a node with missing attributes, the result becomes $v_1$, $v_2$ or $v_6$, and the subtree consists of $\{v_1, v_2, v_6\}$, with a total distance of 2.

To handle the missing information, one simple idea is to first utilize some state-of-the-art graph completion models (such as SAT [6]) to predict the missing information and then apply the existing algorithms (such as BLINKS [2]) to find the answers from the graph with predicted keywords and edges. However, such a completed graph contains many noises and errors comparing with the original graph, and thus this method has poor performance as shown in our experimental study. To capture the latent information of the incomplete graphs, we propose to utilize the graph neural network (GNN) for graph keyword search. GNN has been widely applied in tasks such as link prediction [7, 8, 9], node classification [10, 11], and node clustering [6, 12], but existing models cannot be directly applied to keyword search since they usually embed all the features (keywords) of a node into a single vector and they cannot obtain the representation for the individual query keywords.

We firstly design two naïve approaches based on GNN and dimension reduction. To achieve better performance, we propose a novel auto-encoder and GNN-based model using the message passing mechanism, called KS-GNN. The model mainly consists of three components: an encoder that transforms the original keyword information to low-dimensional embedding vectors; a decoder that aims to reconstruct the high-dimensional representation of keywords from the embedding; a message passing-based aggregation mechanism that preserves the shortest path information between keywords and the target node. Different from the existing graph keyword search works, we propose to leverage GNN to obtain representative node embedding that contains the keyword information, taking the latent graph structure, keyword distribution, and keyword frequency information into account. Meantime, the proposed KS-GNN is able to encode the input query as a low-dimensional vector by its learned powerful encoder, and the results are obtained by computing the similarity between the query embedding and node embeddings. This also speeds up query processing time to linear complexity.

The main contributions of our approach are as follows:

- To our best knowledge, this is the first work on keyword search in graphs with missing information.

- We propose an auto-encoder and GNN-based model KS-GNN to solve the problem effectively without having to know the complete information of the input graph.

- The experimental results on four real-world datasets show that our proposed model consistently outperforms several baseline methods.

## 2   Related Work

**Keyword Search in Graphs.**    Keyword search over graph data aims to find the top-$k$ subtrees or subgraphs according to some ranking criteria. The conventional methods design algorithms assuming that the graphs have complete information. For example, DBXplorer [13] proposes to utilize the number of the answer's edges as the scoring function. BANKS [14] model tuples as nodes in a graph and then performs keyword search using proximity-based ranking. He et al. [2] propose a general ranking function considering both graph structure and content. BLINKS also builds an efficient bi-level index structure to improve efficiency. Kargar and An, motivated by the Steiner tree problem,

use the total edge weight in ranking answers [15]. There also exists studies on keyword search in temporal graphs [16], uncertain graphs [17], knowledge graphs [18], RDF graphs [19], etc. However, real-world graphs may usually be incomplete. As all the state-of-art keyword search methods retrieve the exact answer, the missing information (keywords or edges) imposes a significant effect on the query results. To address this issue, we propose a graph representation learning-based solution to solve the top-$k$ keyword search problem on incomplete graphs.

**Graph Neural Networks.** As a powerful branch of graph representation learning methods, the graph neural network has been widely used in recent years due to its excellent performance. The models in the early years are usually based on the so-called graph convolutional network (GCN) [20, 21, 22], which is based on the Fourier transform theory of graphs developed by Shuman et al. [23]. However, research in recent years has shown that the GCN-based methods can be represented by the message passing mechanism, which is more consistent with the experimental results [24]. The Graph Attention Network (GAT) [11] is one of the representatives of graph neural networks based on the message passing mechanism. GAT introduces the attention mechanism to calculate the attention coefficient between nodes and then uses it to assign different weights to neighbors' information. Based on the auto-encoder and GCN, Graph auto-encoder (GAE) [25] is proposed to reconstruct the adjacency matrix. Moreover, there are some GNN-based works that aim to predict and impute missing data to a data matrix [6, 26, 27]. However, all the methods mentioned above cannot directly handle the graph keyword search problem. To our best knowledge, this is the first work that leverages GNN to process keyword search on incomplete graphs.

## 3 Problem Statement

A graph keyword search query $q = (w_{q_1}, w_{q_2}, ..., w_{q_m})$ contains a set of query keywords, and it searches relevant results from a graph[2] $G = (\mathcal{V}, \mathcal{E}, \mathcal{W})$, where each node $v \in \mathcal{V}$, each edge $e \in \mathcal{E}$, and each keyword $w \in \mathcal{W}$. For each node $v$, it is associated with a set of keywords $\{w_1^v, w_2^v, ..., w_n^v\}$. In this work, we study the keyword search problem over an incomplete graph. To alleviate the effect of the missing information to keyword search over incomplete graphs, we assume that in the original graph the query results are obtained by applying the BLINKS scheme [2] (a commonly used graph keyword search method). Given a query $q$, let $s(v, q)$ denote the score of the node $v$. According to [2, 3, 28, 29, 30], $s(v, q) = \sum_{i=1}^{m} dist_{min}(v, w_{q_i})$, where $dist_{min}(v, w_{q_i})$ computes the shortest path distance from node $v$ to a node containing $w_{q_i}$. BLINKS aims to find top-$k$ nodes where each node can reach all query keywords in the graph, and the scores of the $k$ nodes measured by $s(v, q)$ are the smallest. E.g., in Figure 1(a), $s(v_4, q) = dist_{min}(v_4, c) + dist_{min}(v_4, e) + dist_{min}(v_4, f) = 1$.

**Problem Definition.** Given an incomplete graph $G' = (\mathcal{V}, \mathcal{E}', \mathcal{W}', r_w, r_e)$, where $\mathcal{E}' \subseteq \mathcal{E}, \mathcal{W}' \subseteq \mathcal{W}$, and the proportions of nodes with missing keywords and of missing edges in $G$ are denoted by $r_w$ and $r_e$, respectively. Given a query $q$, the incomplete graph top-$k$ keyword search problem aims to find a set $S = (v_1, v_2, ..., v_k)$ of $k$ nodes from $G'$ such that for any vertex $v' \notin S$, $s(v', q) \geq max(\{s(v_i, q)|v_i \in S\})$.

## 4 Proposed Methods

We propose to solve the keyword search problem with an unsupervised graph representation learning method, since the representative low-dimensional node embeddings can capture the latent information of the input incomplete graph and thus can help recovering the missing information. In addition, low dimensional node embeddings can speed up the query processing by comparing the node embedding with the generated query embedding at the cost of linear complexity. In this section, we first propose two naïve methods that are based on GNN and dimensionality reduction methods, and then we introduce our proposed KS-GNN in details.

### 4.1 Naïve Methods

**Conv-OH.** Graph convolutional layer has been widely used in GNNs, which enables GNN models to gather information from neighbor nodes. Our first naïve method Conv-OH utilizes the graph

---

[2]For ease of presentation, we focus on the undirected graphs, and it is easy to extend the proposed method in directed graphs by passing messages along the edges.

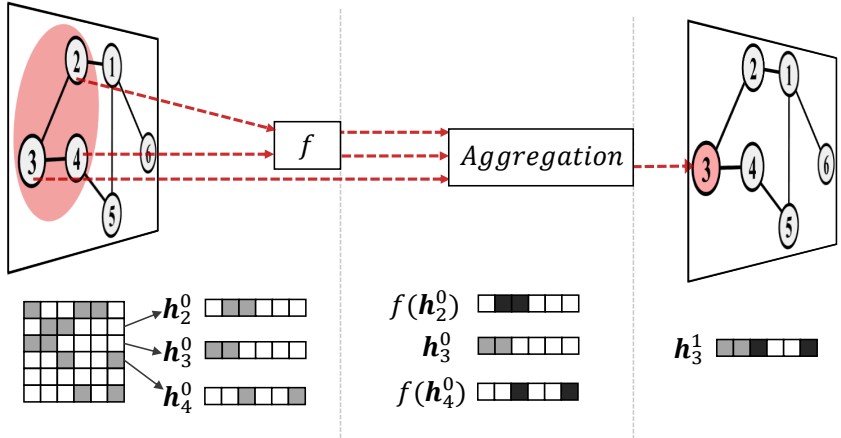

Figure 2: An illustration of the message passing and aggregation of Conv-OH, where $v_3$ is a target node and it aggregates keyword information from its neighbors.

convolutional layer and takes the one-hot encoding of keywords as the input feature for nodes due to the scoring function mentioned in Section 3. Conv-OH is able to return the same answer as does BLINKS [2], if the graph has no missing information and $l$ is large enough.

Frg. 2 shows the message passing and aggregation of Conv-OH. Specifically, with $|\mathcal{V}| = N$ and $|\mathcal{W}| = M$, the one-hot encoding of node $v$ is denoted by $\mathbf{x} = \{0,1\}^M$ with $h_{v,j} = 1$ if $w_j$ is a keyword associated with $v$ and 0 otherwise. Therefore, the input feature matrix is denoted by $\mathbf{X} \in \{0,1\}^{N \times M}$. Let $\mathbf{H}^l$ denote the output node embedding of the $l$-th layer and $\mathbf{h}_v^l$ denote the $v$-th row of $\mathbf{H}^l$, we have:

$$\mathbf{h}_v^{l+1} = Aggregate\Big(\{f(\mathbf{h}_u^l), \forall u \in \mathcal{N}(v)\} \cup \{\mathbf{h}_v^l\}\Big), \tag{1}$$

where $\mathbf{H}^0 = \mathbf{X}$, $f(\cdot)$ denotes a transform function and $\mathcal{N}(v)$ denotes the neighbors of node $v$. Using the combined distance as the scoring function (described in Section 3), Eq. (1) can be written as:

$$\mathbf{h}_v^{l+1} = \Omega\Big(\{\mathbb{1}(\mathbf{h}_u^l) \circ (\mathbf{h}_u^l + \mathbf{1}), \forall u \in \mathcal{N}(v)\} \cup \{\mathbf{h}_v^l\}\Big), \tag{2}$$

where $\Omega(\cdot)$ denotes the element-wise minimum function which ignores zeros, $\circ$ denotes the element-wise product, and $\mathbb{1}(\cdot)$ denotes the indicator function that the element of $\mathbb{1}(\mathbf{h})$ is 1 if the input element is positive and 0 otherwise. For instance, $\Omega(\{[0,0,1,1],[2,0,0,2]\}) = [2,0,1,1]$, and $\mathbb{1}([0,2,0,3]) = [0,1,0,1]$. With Eq. (2), the output of Conv-OH is an $N \times M$ matrix, denoted by $\mathbf{Z}$. Note that there is no dimensionality reduction in Conv-OH and thus this method consumes huge space. The node embedding $\mathbf{h}_v^l$ of $v$ also represents the shortest path distances between the keywords and $v$. Specifically, if $h_{v,i}^l > 0$, it means that the shortest path distance between $v$ and $w_i$ is $h_{v,i}^l - 1$, and $v$ cannot reach $w_i$ within $l$ hops if $h_{v,i}^l = 0$.

For the query processing, given $q$, we can obtain the one-hot encoding of $q$, denoted by $\mathbf{x}_q$. Therefore, given the output node embedding $\mathbf{Z}$, the sum of graph shortest-path distances between nodes and the query keywords can be computed with $\mathbf{x}_q \mathbf{Z}^\top$, and the space complexity is $O(NM)$. It is obvious that Conv-OH cannot deal with the missing information, but it provides some hints to propose more advanced methods.

**Conv-PCA.** Principal component analysis (PCA) is a classic dimensionality reduction technique in multivariate statistical analysis [31]. In order to facilitate data storage and query processing, we propose another naïve PCA-based method to solve the keyword search problem.

Given the one-hot encoding matrix $\mathbf{X}$ as the input feature matrix, PCA is able to keep $d$ principal components of $\mathbf{X}$ with $\mathbf{X}_p = \mathbf{X}\mathbf{U}^\top$, where the rows of $\mathbf{U} \in \mathbb{R}^{d \times M}$ form an orthogonal basis for the $d$ features that are decorrelated [32]. It is worth noting that we can obtain the reconstructed feature matrix $\mathbf{X}'$ with $\mathbf{X}' = \mathbf{X}_p \mathbf{U}$. The learning object is to minimize $\mathcal{L}_{pca} = ||\mathbf{X}' - \mathbf{X}||_2^2$, where $|| \cdot ||_2$ denotes the $L^2$ norm.

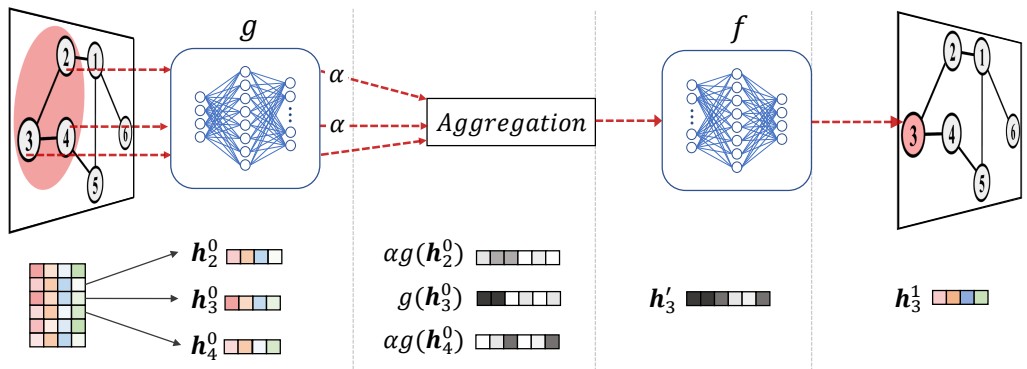

Figure 3: An illustration of the message passing and aggregation of our KS-GNN model.

Inspired by the graph convolutional layers of Conv-OH, we propose **Conv-PCA** with the convolutional layers as below:

$$\mathbf{h}_v^{l+1} = max\Big(\{\alpha\mathbf{h}_u^l, \forall u \in \mathcal{N}(v)\} \cup \{\mathbf{h}_v^l\}\Big), \tag{3}$$

where $\alpha \in (0,1)$ is a decay parameter used to estimate the shortest path distances in Eq. (3), since the dimension is reduced from $M$ to $d$ and thus it is difficult to discriminate $M$ keywords within the $d$ dimensions ($d \ll M$). Specifically, a larger cumulative decay corresponds to a larger shortest path distance. Conv-PCA takes $\mathbf{X}_p$ as the initial node embedding ($\mathbf{H}^0 = \mathbf{X}_p$). This mechanism performs better than directly using PCA in experiments as shown in Section 5.3. For the query processing, given $q$, we can obtain the query embedding $h_q = \mathbf{x}_q\mathbf{U}^\top$. Therefore, given the output node embedding $\mathbf{Z}$, the similarity scores between nodes and the query keywords can be computed by $\mathbf{h}_q\mathbf{Z}^\top$ with linear space complexity $O(dN)$, where the dimension $d$ is a small constant.

Compared with Conv-PCA, Conv-OH utilizes each element of $h_v$ to record the shortest path distance between keywords and $v$ and cannot reduce the dimension of node embedding. Conv-PCA can more efficiently process the keyword search query and requires less space than Conv-OH, but neither of them can well handle the missing information in incomplete graphs.

### 4.2 KS-GNN

Based on the prior discussions on Conv-OH and Conv-PCA, we present the desiderata that guide the development of our method for tackling keyword search as follows:

***Dimensionality Reduction.*** Taking the one-hot encoding matrix $\mathbf{X}$ as input, it is difficult to afford the cost of generating an output with size $N \times M$. Therefore, the model should be able to reduce the dimensions of the output node embedding $M$ to a lower level.

***Key Information Preservation.*** Some keywords and edges information may be lost in the process of dimensionality reduction, which affects the performance of keyword search. The model should retain as much key information as possible to guarantee results quality.

***Adaptive Encoding.*** When generating node embedding, the model should consider the structure information of the target node centered subgraph and the distribution of keywords on the subgraph, rather than only considering the keywords of the target node. Specifically, to recover the missing keywords information in the incomplete graph, the model should be able to capture latent relationships among different keywords. For instance, a pair of keywords "AI" and "ML" often co-occur on nodes near to each other (e.g., one-hop neighbors). Given a node containing either "AI" or "ML", it is natural to assume that the neighbor of this node is more likely to contain the other keyword than the nodes whose one-hop neighbors do not contain either of the two keywords.

***Keyword Frequency Awareness.*** Based on the scoring function in Section 3, the returned top-$k$ nodes tend to be decided by the query keywords with low frequency compared to the high-frequency ones. Thus, for a given keyword, the number of nodes containing it (we denote this by the keywords' *node frequency*) can reflect its importance to the query processing, similar to the inverse document frequency (IDF) used in information retrieval. The keyword set of the whole graph can be regarded as a corpus and the keyword set of each node can be regarded as a document. Therefore, the model

should encode keywords taking in consideration of their frequencies, which are measured based on the keyword node frequency in the whole graph[3].

Based on these desiderata, we propose an auto-encoder based Keyword Search Graph Neural Network (**KS-GNN**) for tackling the problem in incomplete graphs. An illustration of the message passing and aggregation mechanism for generating the node embedding $\mathbf{h}_3$ with KS-GNN is provided in Fig. 3.

**Encoder and Decoder.** KS-GNN employs an encoder $f$ to generate low-dimensional node embedding for *dimensionality reduction*. Recall that in Conv-PCA, the dimension reduction caused information loss and it is hard to discriminate keywords in the low-dimensional space. To address this issue, for the sake of *key information preservation*, KS-GNN employs another decoder $g$ which aims to reconstruct the input from embedding space. By training $f$ simultaneously with $g$, the output embedding of $f$ is able to preserve key information of the input graph. Given the one-hot encoding matrix $\mathbf{X}$ as the input, we define $\mathbf{H} = f(\mathbf{X})$ where $\mathbf{H} \in R^{N \times d}$. For the decoder, it is defined as $\mathbf{X}' = g(\mathbf{H})$ where $\mathbf{X}' \in R^{N \times M}$.

In this work, we utilise the multi-layer perceptron (MLP) with a nonlinear activation layer [33] as both non-linear encoder and decoder. It is worth noting that MLP can be replaced by a more complex neural network. Our goal here is to use simple encoder and decoder to show the advantages of the proposed mechanism. In addition, as a conventional learning objective of the auto-encoder, $f$ and $g$ are trained to minimize:

$$\mathcal{L}_1 = \frac{1}{N}||\mathbf{X}' - \mathbf{X}||_2^2. \tag{4}$$

It is worth noting that the representation PCA learns is essentially the same as that learned by a basic linear auto-encoder. However, the encoder $f$ and the decoder $g$ here are not necessarily linear functions and thus are able to learn more representative node embeddings than the linear ones.

**Message Passing and Aggregation.** The message passing and aggregation mechanisms for both Conv-OH and Conv-PCA are based on the orthogonal basis and decorrelated features. However, in KS-GNN, its encoder $f$ transforms the input without any basis, and thus we cannot simply apply the $\max(\cdot)$ function to capture the information of the nearest keywords on a node. Thanks to the reconstruction ability of the decoder $g$, we can utilise $g$ to reconstruct the $M$-dimension encoding during the message passing and then using $\max(\cdot)$ at this step. Moreover, if the learned node embedding contains the latent information of the missing keyword of the incomplete graph, $g$ can also help recover the missing keywords during the reconstruction. Formally, given the output node embedding $\mathbf{H}^l$ of $l$-th layer, we have:

$$\mathbf{h}_v^{l+1} = f\Big(max\Big(\{\alpha g(\mathbf{h}_u^l), \forall u \in \mathcal{N}(v)\} \cup \{g(\mathbf{h}_v^l)\}\Big)\Big), \tag{5}$$

where $\alpha \in (0,1)$ is a decay parameter that is the same as the one in Eq. (3). As Eq. (5) shows, the $M$-dimension embedding will only be generated by the decoder $g$ during the message passing and aggregation, while the hidden node embedding and the final output node embedding are both $d$ dimensions. Therefore, it meets the requirement of *dimensionality reduction*. The message passing and aggregation can be processed in parallel, and the time-complexity is acceptable. It is worth noting that in incomplete graphs, due to the mechanism of message passing and aggregation, nodes without any keyword information can still be embedded accordingly.

**Subgraph keywords-based Node Similarity.** To realize *adaptive encoding*, we propose to train KS-GNN by a triplet siamese network [34] with a triplet loss [35] according to the subgraph keywords-based node similarity, which enables KS-GNN to capture the latent missing keyword and edge information on incomplete graphs. For a node $v$, we consider the subgraph $SG_v$ containing all the neighbors of $v$ within $k$ hops for measuring node similarity. The one-hot encoding of this subgraph $SG_v$ is denoted by $\mathbf{x}_{SG_v}$. For instance in $G'$ shown in Fig. 1, given $k = 1$, the 1-hop subgraph of $v_5$ contains keywords $\{a, c, d, e, f\}$ with the corresponding one-hot encoding $\mathbf{x}_{SG_5'} = (1, 0, 1, 1, 1, 1)$. Similarly, the one-hot encoding of the subgraphs around $v_4$ and $v_6$ are $\mathbf{x}_{SG_4'} = (1, 1, 1, 0, 0, 1)$ and $\mathbf{x}_{SG_6'} = (1, 1, 1, 1, 1, 1)$, respectively. Therefore, by counting the number of common keywords as the similarity scoring function, we can compare the similarity between $(v5, v4)$ and $(v5, v6)$ by

---

[3]The multi-occurrence of a keyword on one node only contribute 1 to this keyword's node frequency.

comparing $\mathbf{x}_{SG'_5}\mathbf{x}^\top_{SG'_5} = 3$ and $\mathbf{x}_{SG'_5}\mathbf{x}^\top_{SG'_6} = 5$. Specifically, $\mathbf{x}_{SG'_5}\mathbf{x}^\top_{SG'_4} < \mathbf{x}_{SG'_5}\mathbf{x}^\top_{SG'_6}$ indicates $(v5, v6)$ are more similar than $(v5, v4)$ in $G'$.

Given $G'$, the KS-GNN model denoted by $\phi$, and a sampled batch of triplets $\mathcal{T} = \{t_1, t_2, ...t_n\} = \{(v_{o1}, v_{p1}, v_{q1}), (v_{o2}, v_{p2}, v_{q2}), ..., (v_{on}, v_{pn}, v_{qn})\}$, KS-GNN is trained to minimise:

$$\mathcal{L}_2 = \frac{1}{|\mathcal{T}|} \sum_{t_i \in \mathcal{T}} max\left(m - sgn(t_i)\left(\phi(\mathbf{X})_{o_i}\phi(\mathbf{X})^\top_{p_i} - \phi(\mathbf{X})_{o_i}\phi(\mathbf{X})^\top_{q_i}\right), 0\right), \tag{6}$$

where $m$ is a margin hyper-parameter of the hinge loss, $sgn(\cdot)$ denotes a sign function that $sgn(t_i)$ returns 1 if $(v_{oi}, v_{pi})$ are more similar than $(v_{oi}, v_{qi})$ and $-1$ otherwise. Thus, KS-GNN can learn the structure and keyword information from the subgraphs involved.

For large-scale datasets, it might be time-consuming to compute $sgn(t_i)$. In this case, it is acceptable to intuitively sample $\mathcal{T}$ based on the links. For example, for a sampled node $v_{o_i}$, $v_{p_i}$ can be sampled from the 1-hop neighbors of $v_{o_i}$, and $v_{q_i}$ can be negatively sampled from unconnected nodes of $v_{o_i}$, thereby setting $sgn(t_i)$ to 1. Eq. (6) still takes both the graph structural information and the keyword distribution into account by feeding the one-hot encoding of subgraph keywords to KS-GNN. In addition, minimizing Eq. (6) helps generate similar adaptive embedding for the keywords which co-occur commonly.

**Keyword Frequency-based Regularization.** Intuitively, if a keyword appears on many nodes, it is regarded as less important than the keyword which appears on fewer nodes for query processing. Therefore, in this work, we consider the keyword node frequency, denoted by $c_i$, that indicates the number of nodes containing keyword $w_i$. For instance, in $G'$ shown in Fig. 1, $c_1 = 2$ and $c_2 = 2$ for keywords $a$ and $b$, respectively. We propose to enhance the model's *keyword frequency awareness* with a regularization that minimizes:

$$\mathcal{L}_3 = \frac{1}{M} \sum_{w_i \in \mathcal{W}} c_i ||f(\mathbf{I}_i)||_2, \tag{7}$$

where $\mathbf{I}$ denotes an $M \times M$ identity matrix, and $\mathbf{I}_i$ denotes the $i$-th row of $\mathbf{I}$. Feeding $\mathbf{I}_i$ in $f$ can return the representation of keyword $w_i$, and minimizing Eq. (7) aims to differentiate the lengths of keyword embeddings according to their keyword frequencies, thereby being aware of keyword frequency.

To train KS-GNN, the final **learning objective** is to minimize:

$$\mathcal{L} = \lambda_1\mathcal{L}_1 + \lambda_2\mathcal{L}_2 + \lambda_3\mathcal{L}_3, \tag{8}$$

where $\lambda_1$, $\lambda_2$ and $\lambda_3$ are hyper-parameters. By minimizing Eq. (8), we can optimize KS-GNN to generate informative node embedding which can capture the latent representation of missing keywords and edges. The superiority of the proposed KS-GNN is validated in Section 5.3.

**Query Processing.** To process query $q$, given the one-hot encoding of $q$ as $\mathbf{x}_q$, the trained encoder $f$ and the learned node embedding $\mathbf{Z}$, we can compute the similarity between the nodes and query with $\mathbf{s_q} = f(\mathbf{x}_q)\mathbf{Z}^\top$, and the top-k answers can be found with the largest scores in $\mathbf{s_q}$. In addition, the space complexity of computing query processing is $O(dN)$.

# 5 Experiments

In this section, we evaluate the performance of our proposed approach, KS-GNN, on four real-world datasets, including citation networks (**CiteSeer**), co-purchase networks (**Video & Toy**) and co-author networks (**DBLP**). The details of datasets, additional experimental results and analysis can be found in the supplementary materials.

## 5.1 Baseline Methods

We compare our model against five baseline methods, including a state-of-the-art deep learning based missing-data completion GNN model. More details on the baseline models are provided in the supplementary materials.

- **GraphSAGE** [10] is a representative GNN-based graph embedding method. We add an MLP encoder for GraphSAGE to address the keyword search problem for GraphSAGE.

- **BLINK+SAT** firstly predicts and completes the missing keywords and edges with a state-of-the-art missing-data completion GNN model SAT [6] and then utilises BLINK [2] to process keyword search on the new graph.

- **PCA** is based on the classic dimensionality reduction technique [31].

- **Conv-PCA** is a naïve method proposed in Section 4.1.

- **KS-PCA** is a simplified variant of our proposed KS-GNN that replaces the MLP encoder and decoder with PCA transformations. Specifically, it leverages $\mathbf{U}$ to reconstruct $M$-dimension embedding from $\mathbf{h}_v$.

## 5.2 Experimental Setup

In our experiments, we compare the proposed method with baseline methods for keyword search tasks in two kinds of graphs: (1) the graphs with only missing keywords; (2) the graphs with both missing keywords and edges. For each dataset, to simulate a real-world scenario and quantitatively control the ratios of missing information, we process the original datasets with two steps: (1) hide the keywords of randomly sampled nodes with a predefined proportion (denoted by $r_w$) in the graph; (2) randomly hide a proportion (denoted by $r_e$) of the edges in the graph. Let $n_q = |q|$ denote the number of words in the query $q$, we randomly sample **100** queries as the test set for each value of $n_q$ ranging from 3 to 9 with a step of 2.

In addition, in each incomplete graph, the validation set consists of 100 randomly generated queries with ground truth answers. We tune the hyper-parameters of compared methods with the grid search algorithm on the validation set, more details can be found in the supplementary materials. In terms of the evaluation metric, we use Hits@$K$, which is a common ranking metric that counts the ratio of positive edges that are ranked at the $K$-th place or above. The ground truth is the top-$K$ answers retrieved by BLINK on the original graph for each dataset. Specifically, we report Hits@100, and more experimental results (Hits@10 and Hits@50) can be found in the appendix.

## 5.3 Performance of Keyword Search

Table 1 shows the comparison results in graphs with $r_w$ adjusted from 0.3 to 0.7 and $n_q$ adjusted from 3 to 9. As shown in the table, KS-GNN and KS-PCA significantly outperforms the baselines, and changing $r_w$ will not affect its performance. Moreover, the performance of KS-GNN better when more query keywords are given. As for the baselines, BLINK+SAT cannot maintain good performance when many keywords are missing. Compared with Conv-PCA, KS-PCA can address the keyword search in incomplete graphs much more effectively due to the proposed novel message passing and aggregation mechanism. Although KS-PCA and KS-GNN have similar message-passing mechanisms, KS-GNN performs better in most cases due to its more representative output node embeddings. This reveals that the proposed learning objective and auto-encoder-based model are able to enhance the ability of representation learning. Since PCA focuses on each single node, it performs well when the query keywords are located on the same node. However, when $n_q$ increases, the query keywords tend to be located on different nodes, and the performance of PCA therefore decreases because it cannot gather neighbor information. By contrast, although GraphSAGE can aggregate information from neighbors, it sometimes performs worse than PCA, because only utilizing the max-pooling operator during the message aggregation cannot well distinguish the information from each unique keyword. This can be proved by that Conv-PCA performs better than both PCA and GraphSAGE, which also indicates the superiority of our proposed encoder and decoder-based message passing and aggregation mechanism.

Table 2 presents the results of the comparison in graphs with both missing keywords and edges, where $r_e$ is set to 0.3 and $r_w$ is adjusted from 0.3 to 0.7. As the table shows, KS-GNN still outperforms other compared baseline methods in most cases since it can learn the adaptive embedding and structural information from the incomplete graph with missing keywords and edges. Compared with Table 1, Table 2 shows that the effect of missing edges cannot be reflected for the powerless methods, such as GraphSAGE and BLINK+SAT, because they might gather noisy information from neighbours. For KS-GNN, the effect of missing edges varies on different datasets. For instance, KS-GNN is more robust on CiteSeer and DBLP than on Video and Toy.

Table 1: Method performance by Hits@100 (%) in graphs with only missing keywords.

| Datasets | $r_w$ | 0.3 | | | | 0.5 | | | | 0.7 | | | |
|---|---|---|---|---|---|---|---|---|---|---|---|---|---|
| | $n_q$ | 3 | 5 | 7 | 9 | 3 | 5 | 7 | 9 | 3 | 5 | 7 | 9 |
| CiteSeer | GraphSAGE | 6.87 | 5.64 | 2.95 | 3.71 | 5.15 | 4.74 | 3.52 | 2.07 | 9.42 | 8.27 | 6.05 | 7.54 |
| | BLINK+SAT | 8.81 | 7.23 | 5.65 | 4.42 | 9.14 | 5.97 | 5.47 | 4.88 | 9.82 | 8.34 | 8.75 | 5.49 |
| | PCA | 10.26 | 7.73 | 6.51 | 5.19 | 9.31 | 6.70 | 6.28 | 4.82 | 7.73 | 6.40 | 6.47 | 4.98 |
| | Conv-PCA | 8.49 | 8.43 | 8.76 | 7.41 | 8.92 | 7.78 | 11.28 | 13.58 | 11.38 | 9.20 | 11.44 | 9.90 |
| | **KS-PCA** | 24.91 | 27.92 | 33.15 | 38.35 | 23.61 | 25.87 | 32.84 | 30.74 | 20.94 | 26.18 | 35.59 | 33.99 |
| | **KS-GNN** | **30.84** | **37.86** | **38.07** | **42.61** | **31.43** | **38.79** | **38.86** | **42.62** | **28.69** | **35.25** | **35.61** | **38.64** |
| Video | GraphSAGE | 0.49 | 0.30 | 0.06 | 0.03 | 0.44 | 0.14 | 0.00 | 0.05 | 0.34 | 0.35 | 0.21 | 0.16 |
| | BLINK+SAT | 10.21 | 9.86 | 10.99 | 14.87 | 8.55 | 6.92 | 8.63 | 5.82 | 1.18 | 1.15 | 4.38 | 3.35 |
| | PCA | 1.54 | 0.91 | 0.55 | 0.61 | 1.71 | 0.72 | 0.71 | 0.57 | 1.66 | 0.95 | 0.66 | 0.55 |
| | Conv-PCA | 1.81 | 2.46 | 1.58 | 2.38 | 2.43 | 1.49 | 1.66 | 2.54 | 2.42 | 2.37 | 2.80 | 3.37 |
| | **KS-PCA** | 10.19 | 12.23 | 16.15 | 21.37 | 11.26 | 15.62 | 19.51 | 23.87 | 10.66 | 15.57 | 19.17 | 25.36 |
| | **KS-GNN** | **21.43** | **23.36** | **22.92** | **26.79** | **22.54** | **22.57** | **30.41** | **33.41** | **21.01** | **16.48** | **22.01** | **28.47** |
| Toy | GraphSAGE | 0.74 | 0.88 | 5.11 | 4.21 | 4.24 | 12.49 | 6.50 | 6.10 | 1.45 | 4.39 | 6.71 | 0.09 |
| | BLINK+SAT | 6.47 | 8.17 | 8.12 | 10.54 | 6.79 | 9.59 | 10.99 | 11.93 | 6.44 | 8.56 | 11.55 | 8.15 |
| | PCA | 1.15 | 0.68 | 0.63 | 0.51 | 1.01 | 0.69 | 0.51 | 0.44 | 0.67 | 0.47 | 0.44 | 0.30 |
| | Conv-PCA | 21.34 | 21.99 | 23.76 | 25.40 | 19.17 | 19.72 | 20.21 | 25.22 | 16.99 | 21.61 | 23.95 | 24.90 |
| | **KS-PCA** | 27.23 | 27.73 | **31.58** | 33.79 | **25.78** | 28.94 | 31.04 | 32.82 | 18.35 | 22.03 | 25.50 | 26.25 |
| | **KS-GNN** | **28.56** | **29.85** | 29.55 | **34.28** | 24.65 | **29.16** | **31.27** | **33.25** | **21.78** | **27.41** | **25.55** | **30.17** |
| DBLP | GraphSAGE | 0.42 | 0.49 | 0.36 | 0.82 | 0.29 | 0.43 | 0.53 | 0.05 | 0.07 | 0.01 | 0.01 | 0.00 |
| | BLINK+SAT | 3.26 | 6.09 | 4.01 | 6.65 | 3.49 | 1.66 | 5.42 | 3.95 | 4.25 | 2.42 | 3.12 | 4.24 |
| | PCA | 3.78 | 2.57 | 2.25 | 2.38 | 3.06 | 2.55 | 2.21 | 2.15 | 2.97 | 2.51 | 2.02 | 1.88 |
| | Conv-PCA | 9.00 | 13.24 | 13.29 | 16.87 | 5.93 | 6.56 | 7.64 | 10.62 | 7.00 | 10.52 | 10.03 | 15.68 |
| | **KS-PCA** | 15.28 | 21.41 | 25.61 | 31.64 | 14.98 | 20.73 | 23.21 | **31.72** | 12.49 | 19.49 | 21.23 | 28.63 |
| | **KS-GNN** | **16.21** | **24.94** | **29.55** | **33.51** | **16.52** | **22.73** | **26.85** | 30.69 | **15.57** | **24.15** | **27.12** | **29.06** |

Table 2: Method performance by Hits@100 (%) in graphs with both missing keywords and edges ($r_e$ = 0.3).

| Datasets | $r_w$ | 0.3 | | | | 0.5 | | | | 0.7 | | | |
|---|---|---|---|---|---|---|---|---|---|---|---|---|---|
| | $n_q$ | 3 | 5 | 7 | 9 | 3 | 5 | 7 | 9 | 3 | 5 | 7 | 9 |
| CiteSeer | GraphSAGE | 2.04 | 3.22 | 2.99 | 1.72 | 3.89 | 2.95 | 1.03 | 1.11 | 9.62 | 8.26 | 7.35 | 7.23 |
| | BLINK+SAT | 9.19 | 8.14 | 5.68 | 3.71 | 8.26 | 6.79 | 6.04 | 8.57 | 7.88 | 8.33 | 7.69 | 8.30 |
| | PCA | 10.26 | 7.73 | 6.51 | 5.19 | 9.31 | 6.70 | 6.28 | 4.82 | 7.73 | 6.40 | 6.47 | 4.98 |
| | Conv-PCA | 9.61 | 9.37 | 8.17 | 7.68 | 10.82 | 9.22 | 11.83 | 14.33 | 11.40 | 10.17 | 11.91 | 9.94 |
| | **KS-PCA** | 25.27 | 27.89 | 31.47 | 35.71 | 22.51 | 24.92 | 31.65 | 30.49 | 20.72 | 25.60 | 35.30 | 32.81 |
| | **KS-GNN** | **30.57** | **37.88** | **38.15** | **41.80** | **26.80** | **34.70** | **34.37** | **36.75** | **24.47** | **31.19** | **35.82** | **34.96** |
| Video | GraphSAGE | 0.09 | 0.11 | 0.02 | 0.00 | 0.26 | 0.05 | 0.00 | 0.04 | 1.65 | 1.65 | 2.22 | 1.29 |
| | BLINK+SAT | 1.67 | 1.85 | 2.48 | 1.44 | 0.08 | 0.99 | 4.96 | 2.97 | 2.19 | 1.77 | 0.78 | 1.21 |
| | PCA | 1.54 | 0.91 | 0.55 | 0.61 | 1.71 | 0.72 | 0.71 | 0.57 | 1.66 | 0.95 | 0.66 | 0.55 |
| | Conv-PCA | 1.05 | 1.59 | 0.83 | 2.01 | 1.43 | 0.81 | 0.87 | 1.23 | 1.25 | 1.25 | 1.31 | 1.38 |
| | **KS-PCA** | 3.82 | 4.13 | 4.88 | 7.13 | 3.96 | 4.52 | 5.33 | 6.11 | 3.64 | 4.58 | 5.17 | 6.55 |
| | **KS-GNN** | **8.08** | **8.34** | **12.88** | **11.82** | **6.84** | **7.68** | 4.18 | **11.12** | **6.37** | **10.31** | **13.92** | **10.07** |
| Toy | GraphSAGE | 0.04 | 0.02 | 0.01 | 0.00 | 0.28 | 0.00 | 0.01 | 0.02 | 0.02 | 0.18 | 0.29 | 0.23 |
| | BLINK+SAT | 3.69 | 3.03 | 4.85 | 5.66 | 2.56 | 1.34 | 1.86 | 5.46 | 1.76 | 1.39 | 4.73 | 4.45 |
| | PCA | 1.15 | 0.68 | 0.63 | 0.51 | 1.01 | 0.69 | 0.51 | 0.44 | 0.67 | 0.47 | 0.44 | 0.30 |
| | Conv-PCA | 11.64 | 10.46 | 11.23 | 12.29 | 9.31 | 9.00 | 9.08 | 11.61 | 6.74 | 8.87 | 8.99 | 9.56 |
| | **KS-PCA** | 13.80 | 13.03 | 13.55 | **15.67** | 11.35 | 11.03 | 11.09 | **13.02** | 6.84 | 8.37 | 9.50 | 10.43 |
| | **KS-GNN** | **13.82** | **13.28** | **14.38** | 15.22 | **12.51** | **12.54** | **12.68** | 12.59 | **9.41** | **8.99** | **12.83** | **11.15** |
| DBLP | GraphSAGE | 0.51 | 0.20 | 0.42 | 0.34 | 0.67 | 0.25 | 0.40 | 0.90 | 0.40 | 0.05 | 0.04 | 0.02 |
| | BLINK+SAT | 4.78 | 3.45 | 6.74 | 4.58 | 3.94 | 3.66 | 5.05 | 3.97 | 2.96 | 2.75 | 2.83 | 7.29 |
| | PCA | 3.78 | 2.57 | 2.25 | 2.38 | 3.06 | 2.55 | 2.21 | 2.15 | 2.97 | 2.51 | 2.02 | 1.88 |
| | Conv-PCA | 8.35 | 11.8 | 12.51 | 14.51 | 5.25 | 6.77 | 7.03 | 9.91 | 6.46 | 9.59 | 9.47 | 14.33 |
| | **KS-PCA** | 15.05 | 19.99 | 22.61 | 28.57 | 13.68 | 19.19 | 21.59 | 28.96 | 11.67 | **17.71** | 18.38 | **24.51** |
| | **KS-GNN** | **15.91** | **20.49** | **25.09** | **29.04** | **15.79** | **19.86** | **22.15** | **29.71** | **12.07** | 17.18 | **19.23** | 24.19 |

## 5.4 Ablation Study

In this section, we further investigate how each component in Eq. (8) affects KS-GNN's performance. The experiments are conducted in the incomplete graphs with $r_e = 0$ and $r_w = 0.3$ on four datasets. For the queries, the number of keywords is set to $n_q = 3$. Based on Eq. (8), we have three variants which are trained with different learning objectives, denoted by *without* $\mathcal{L}_1$, *without* $\mathcal{L}_2$ and *without* $\mathcal{L}_3$, respectively. For instance, the learning objective is $\lambda_2 \mathcal{L}_2 + \lambda_3 \mathcal{L}_3$ for the variant *without* $\mathcal{L}_1$. The hyper-parameters are still tuned with the grid search algorithm. The results of Hits@100 scores are shown in Table 3. As the table shows, among each component in Eq. (8), the average performance gains for $\mathcal{L}_1$, $\mathcal{L}_2$ and $\mathcal{L}_3$ on four datasets are 12.23%, 6.87% and 10.32%, respectively. The results also show that these three learning objectives play different roles on different datasets.

Table 3: **Hits@100** (%) in the ablation study of KS-GNN.

|  | CiteSeer | Video | Toy | DBLP |
|---|---|---|---|---|
| *without* $\mathcal{L}_1$ | 28.17 | 17.82 | 24.67 | 14.62 |
| *without* $\mathcal{L}_2$ | 29.51 | 18.86 | 26.39 | 15.63 |
| *without* $\mathcal{L}_3$ | 26.28 | 17.13 | 27.01 | 16.05 |
| KS-GNN | **30.84** | **21.43** | **28.56** | **16.21** |

## 6 Conclusion

Keyword search in graphs is an important problem with many applications such as network analysis and recommendation. The keywords and edges in graphs might be lost or incomplete due to some reasons in real-world applications, such as storage limitation or privacy issues. In this paper, we study the keyword search problem in incomplete graphs and propose a novel auto-encoder and GNN-based method, KS-GNN. Compared to existing methods, KS-GNN is able to address the problem when some nodes have missing keywords or some edges are missing in the input graphs. The results of extensive experiments on real-world datasets reveal that KS-GNN significantly outperforms the state-of-the-art baseline methods on the incomplete graph keyword search task.

## Acknowledgments and Disclosure of Funding

Xin Cao is supported by ARC DE190100663. Yixiang Fang is supported by CUHK-SZ grant UDF01002139.

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
