# KS-GNN: Keywords Search over Incomplete Graphs via Graphs Neural Network

**Yu Hao**
University of New South Wales
NSW, Australia
yu.hao@unsw.edu.au

**Xin Cao**[*]
University of New South Wales
NSW, Australia
xin.cao@unsw.edu.au

**Yufan Sheng**
University of New South Wales
NSW, Australia
yufan.sheng@unsw.edu.au

**Yixiang Fang**
Chinese University of Hong
Kong, Shenzhen, China
fangyixiang@cuhk.edu.cn

**Wei Wang**
The Hong Kong University of Science
and Technology, Guangzhou, China
weiwcs@ust.hk

## A    Additional Experimental Details

For PCA-based methods, the dimensionality reduction is performed via singular value decomposition (SVD) of the input one-hot encoding matrix $\mathbf{X}$. As mentioned above, we utilize grid search for tuning the hyper-parameters. In particular, for the learning-based methods, including GraphSAGE and KS-GNN, the learning rates are selected from $\{0.1, 0.01, 0.001, 0.0001\}$. For the convolutional neural networks (i.e. GraphSAGE, SAT, Conv-PCA, KS-PCA, KS-GNN), we swept the number of hidden layers in the set $\{1, 2, 3, 4, 5\}$. For the other hyper-parameters used in KS-GNN, such as $\lambda_1$, $\lambda_2$ and $\lambda_3$, we tune them from 0.1 to 1 with a step of 0.1. As for the margin hyper-parameter $m$ in Eq.(6), we search it from $\{0, 0.1, 0.5, 1, 2.5, 5, 10\}$. For SAT, we follow the hyper-parameter setting in [1], such as tuning $\lambda_{3c}$ from 0.1 to 100. In the conducted experiments, the default hidden dimension is selected from $\{128, 256, 512\}$ according to the result of grid search, while the default dimension of output node embedding is 64.

## B    Datasets

**CiteSeer** [2] is a citation network, where each node represents a document and edge represents a citation [2, 3]. The keywords on each node are extracted by stemming and stop word removal.

**Video & Toy** [3] are two co-purchase networks, which are sampled from Amazon Video Games and Amazon Toys respectively [4]. The nodes represent the products, and the keywords represent the features. Two nodes are connected if they are both purchased by one customer. However, there are some differences between the two datasets. **Video** contains more keywords while **Toy** has more connections.

---

[*]Corresponding author.

[2]https://github.com/kimiyoung/planetoid/raw/master/data

[3]http://deepyeti.ucsd.edu/jianmo/amazon/index.html

35th Conference on Neural Information Processing Systems (NeurIPS 2021).

**DBLP** [4] is a co-author network from DBLP. The nodes represent the researchers, while the keywords associated with each node are extracted from the abstracts of the author's work. The edges represent the co-authorship between the authors.

Table 1 presents detailed overview of the relevant statistics.

Table 1: Statistics of experimental datasets.

| Datasets | #Nodes | #Edges | #Keywords |
|----------|--------|--------|-----------|
| CiteSeer | 3,327 | 9,104 | 3,703 |
| Video | 20,882 | 66,003 | 11,514 |
| Toy | 20,682 | 224,603 | 4,114 |
| DBLP | 32,361 | 69,448 | 4,094 |

## C  Baseline Methods

**GraphSAGE** [5] is a representative GNN-based graph embedding method, which aggregates neighbor information by multiple convolution layers. To address keyword search, GraphSAGE needs an additional encoder before the forward propagation. Therefore, we add an MLP encoder, denoted by $\psi$, for GraphSAGE, and this encoder can also be used to generate query embedding with $\psi(\mathbf{x}_q)$. To be compared with KS-GNN, GraphSAGE employs the max-pooling operator in this work.

**BLINK+SAT** is a combination method, which is based on a conventional keyword search method BLINK [6] and a state-of-the-art missing-data completion GNN model SAT [1]. This baseline method leverages SAT to predict and complete the missing keywords and edges first and then utilises BLINK to process keyword search on the new graph.

**PCA** [7] reduces the dimensions of the input one-hot encoding feature matrix $\mathbf{X}$ from $M$ to $d$ by learning the basis $\mathbf{U}$ with $\mathbf{X}_p = \mathbf{X}\mathbf{U}^\top$. For the query process, given a query $q$, the query embedding $h_q$ equals $\mathbf{x}_q\mathbf{U}^\top$. The query answers are returned according to the similarity scores directly computed by $\mathbf{X}_p\mathbf{x}_q^\top$.

**Conv-PCA** is a naive method proposed in Section 4.1. In addition, based on the discussion of KS-GNN, we further propose a variant of Conv-PCA that leverages $\mathbf{U}$ to reconstruct $M$-dimension embedding from $\mathbf{h}_v$, namely **KS-PCA**. Formally, the aggregation of KS-PCA is:

$$\mathbf{h}_v^{l+1} = max(\{\alpha\mathbf{h}_u^l\mathbf{U}, \forall u \in \mathcal{N}(v)\} \cup \{\mathbf{h}_v^l\mathbf{U}\})\mathbf{U}^\top. \tag{1}$$

The query process of KS-PCA is the same as that of Conv-PCA.

## D  Searching for Subgraph Algorithm

Algorithm 1 is a BFS-based algorithm that searches for a subgraph when given a returned root node $v_r$ by KS-GNN and a query $q$. When checking a node embedding, Algorithm 1 needs a threshold parameter $\sigma$ which can help indicate if the node contains a keyword. Specifically, for a node $v$, the original keyword information can be restored from its node embedding with $g(\mathbf{z}_v) \in \mathbb{R}^M$. For each element in $g(\mathbf{z}_v)$, we assume that the node $v$ contains a word $w_i$ if and only if when $g(\mathbf{z}_v)[i] > \sigma$. Based on this, we can leverage the BFS algorithm to check the neighbors of the root node $v_r$, thereby getting a subgraph when all keywords in the query have been found or all nodes have been visited.

## E  Additional Experimental Results

### E.1  Query Processing Efficiency

As discussed above, our proposed KS-GNN is able to answer the query within the time complexity of $O(dN)$. Figure 1 shows the time efficiency experiments conducted on DBLP dataset. In each

---

[4] https://dblp.uni-trier.de

**Algorithm 1:** Search for Subgraph

---

**Input:** The trained output node embedding $\mathbf{Z}$, query $q = (w_{q_1}, w_{q_2}, ..., w_{q_m})$, root node $v_r$,
      encoder $f$, decoder $g$, threshold $\sigma$
**Output:** The subgraph $SG_r$.
$SG_r := \emptyset$ ;
$S_n := v_r)$;
// initialize $S_n$
$S' := \emptyset$ ;
// the set of nodes having been visited
**while** $q \neq \emptyset$ and $SG_r \neq \emptyset$ **do**
    **for** $w_i \in q$ **do**
        $u, s_{w_i} = IndexMax(\{g(\mathbf{z}_u)[w_i], u \in S_n\})$;
        **if** $s_{w_i} > \sigma$ **then**
            $q := q.delete(w_i)$;
            $SG_r := SG_r.add(u)$;

    **if** $q = \emptyset$ **then**
        break;
    $S'.update(S_n)$;
    $S_n := \{\mathcal{N}(v), v \in S_n\}/S'$;
**return** $SG_r$;

---

experiment, we set $d$ to 64 and change the number of nodes from $10^2$ to $10^7$. The experiments are conducted based on the RTX 2080 Ti GPU and PyTorch. When the nodes in DBLP are not sufficient, we add some synthetic nodes to meet the number requirements. As the figure shows, KS-GNN is able to process the query linearly, and the run-time changes slightly when the number of nodes changes from $10^2$ to $10^7$.

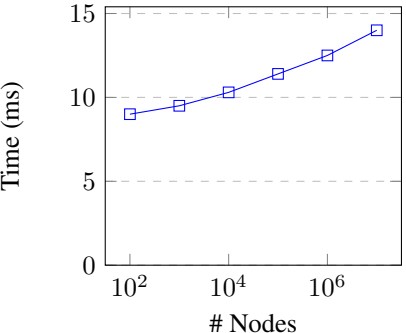

Figure 1: Query processing time in seconds (ms).

## E.2 Performance of Keyword Search

Table 2 and Table 3 shows the results of experiments on keyword search in graphs with only missing keywords for Hits@10 and Hits@50 scores, respectively. From the tables, we can find that the results are consistent with the experimental results shown in Section 5.3, where our proposed KS-GNN significantly outperforms the baseline methods. We further include the variances of the results of KS-GNN varying the seed of edge and keyword sampling as shown in Table 4 and Table 5. These tables show the robustness of our proposed method.

In addition, it is worth noting that answering keyword search queries with high Hits@10 scores is more challenging than that of Hits@100. However, KS-GNN is able to achieve a Hits@10 score of 42.5% on the Video dataset with $r_w = 0.3$ for 9-keyword queries, while the best keyword search performance of other compared methods is achieved by KS-PCA, which is only 7.2% on the same task. It can also be found that GraphSAGE cannot return a good answer when the number of answers

Table 2: Method performance by Hits@10 in graphs with only missing keywords.

| Datasets | $r_w$ | 0.3 | | | | 0.5 | | | | 0.7 | | | |
|---|---|---|---|---|---|---|---|---|---|---|---|---|---|
| | $n_q$ | 3 | 5 | 7 | 9 | 3 | 5 | 7 | 9 | 3 | 5 | 7 | 9 |
| CiteSeer | GraphSAGE | 0.20 | 0.10 | 0.50 | 0.15 | 0.05 | 0.40 | 0.65 | 0.00 | 0.40 | 0.85 | 0.70 | 0.50 |
| | BLINK+SAT | 3.96 | 4.40 | 4.86 | 4.03 | 2.75 | 2.54 | 3.78 | 2.71 | 1.94 | 1.39 | 1.65 | 1.02 |
| | PCA | 6.80 | 3.50 | 2.00 | 1.60 | 5.90 | 2.80 | 1.00 | 1.60 | 7.40 | 3.90 | 3.10 | 2.80 |
| | Conv-PCA | 0.80 | 2.30 | 1.90 | 1.10 | 1.11 | 1.40 | 2.60 | 2.60 | 2.90 | 2.10 | 1.50 | 2.60 |
| | KS-PCA | 8.20 | 7.70 | 10.20 | 14.20 | 5.40 | 8.80 | 9.01 | 10.70 | 6.50 | 7.10 | 9.10 | 11.20 |
| | KS-GNN | 8.30 | 10.80 | 15.19 | 14.95 | 7.72 | 10.68 | 16.68 | 17.48 | 8.11 | 11.27 | 16.19 | 17.87 |
| Video | GraphSAGE | 0.10 | 0.00 | 0.00 | 0.00 | 0.10 | 0.00 | 0.00 | 0.00 | 0.00 | 0.00 | 0.00 | 0.00 |
| | BLINK+SAT | 1.53 | 1.26 | 1.43 | 2.03 | 1.18 | 0.96 | 1.20 | 0.85 | 0.22 | 0.25 | 0.65 | 0.31 |
| | PCA | 0.00 | 0.00 | 0.10 | 0.00 | 0.50 | 0.20 | 0.10 | 0.00 | 1.10 | 0.10 | 0.00 | 0.10 |
| | Conv-PCA | 1.00 | 0.30 | 0.70 | 0.50 | 0.30 | 0.50 | 0.50 | 0.50 | 0.10 | 0.40 | 0.60 | 0.70 |
| | KS-PCA | 2.40 | 5.30 | 7.00 | 7.20 | 1.50 | 4.30 | 7.20 | 8.80 | 2.80 | 5.20 | 9.90 | 13.30 |
| | KS-GNN | 15.30 | 18.90 | 16.70 | 42.50 | 15.40 | 13.80 | 24.20 | 45.10 | 8.30 | 15.10 | 21.39 | 41.20 |
| Toy | GraphSAGE | 0.00 | 0.10 | 0.00 | 0.20 | 0.00 | 0.10 | 0.00 | 0.00 | 0.10 | 0.00 | 0.10 | 0.00 |
| | BLINK+SAT | 2.51 | 2.90 | 1.20 | 2.80 | 1.00 | 1.10 | 2.50 | 0.18 | 1.70 | 1.60 | 1.98 | 0.20 |
| | PCA | 0.30 | 0.20 | 0.10 | 0.00 | 0.70 | 0.30 | 0.10 | 0.00 | 0.30 | 0.00 | 0.00 | 0.12 |
| | Conv-PCA | 3.60 | 3.20 | 3.90 | 5.00 | 2.90 | 2.70 | 3.20 | 4.20 | 1.70 | 1.60 | 3.00 | 2.06 |
| | KS-PCA | 6.10 | 4.90 | 5.30 | 7.60 | 3.50 | 4.70 | 4.30 | **6.70** | 2.70 | 2.40 | 3.10 | 2.73 |
| | KS-GNN | **7.10** | **5.60** | **7.20** | **8.80** | **9.10** | **5.70** | **5.30** | 6.40 | **5.10** | **3.60** | **6.70** | **6.90** |
| DBLP | GraphSAGE | 0.00 | 0.00 | 0.00 | 0.00 | 0.05 | 0.00 | 0.20 | 0.00 | 0.00 | 0.00 | 0.00 | 0.00 |
| | BLINK+SAT | 2.46 | 1.13 | 3.13 | 4.14 | 0.93 | 0.33 | 1.73 | 2.78 | 0.33 | 3.20 | 0.46 | 3.11 |
| | PCA | 1.60 | 1.10 | 1.01 | 0.70 | 1.40 | 1.50 | 0.50 | 0.40 | 1.50 | 0.70 | 0.60 | 0.60 |
| | Conv-PCA | 3.20 | 6.30 | 8.80 | 8.20 | 0.80 | 4.10 | 3.20 | 4.20 | 3.05 | 5.20 | 3.80 | 9.20 |
| | KS-PCA | 5.90 | 11.90 | 19.70 | 23.80 | 4.50 | 8.90 | 18.20 | 22.40 | 4.00 | 8.60 | 15.70 | 19.70 |
| | KS-GNN | **10.21** | **20.56** | **27.78** | **33.36** | **7.89** | **24.50** | **31.82** | **36.06** | **8.13** | **23.42** | **30.82** | **38.22** |

is limited to 10 on the large-scale datasets, such as Video and Toy. PCA is able to achieve better performance than GraphSAGE and Conv-PCA on CiteSeer, and the reason might be the query keywords locate on the same node and Conv-PCA cannot well capture the latent keyword information in a sparse graph like the CiteSeer dataset. In contrast, our proposed KS-GNN is able to generate informative node embedding for handling keyword search problem in incomplete graphs.

### E.3 Sensitivity Analysis of Number of Convolutional Layers

The proposed KS-GNN aggregates the information of neighbors based on the convolution layer, and the amount of aggregated information depends on the number of layers. Therefore, to figure out how the number of convolutional layers affects KS-GNN's performance. Specifically, we conduct experiments of KS-GNN with different layer numbers changing from 1 to 5 on CiteSeer and DBLP, which are small and large datasets, respectively. We set both $r_e$ and $r_w$ to 0.3. For each query, the number of keywords is set to $n_q = 5$. The results are shown in Fig. 2. As the figure shows, the number of layers influences KS-GNN's performance. In general, setting the number of layers to 3 can achieve better performance than others. This also indicates that it is not necessary to make KS-GNN as deep as possible.

### E.4 Impact of $d$

We further conduct experiments on the CiteSeer dataset to investigate the impact of changing the output dimension of node embedding ($d$). To thoroughly observe the impact of $d$, we set the dimension of hidden layers as $512$ and change $d$ from 32 to 512. The results are shown in Fig. 3. As the figure shows, larger $d$ can help KS-GNN increase the performance. However, for other baseline methods, changing $d$ cannot improve their performance significantly, especially for PCA and GraphSAGE.

Table 3: Method performance by Hits@50 in graphs with only missing keywords.

| Datasets | $r_w$ | 0.3 | | | | 0.5 | | | | 0.7 | | | |
|---|---|---|---|---|---|---|---|---|---|---|---|---|---|
| | $n_q$ | 3 | 5 | 7 | 9 | 3 | 5 | 7 | 9 | 3 | 5 | 7 | 9 |
| CiteSeer | GraphSAGE | 2.29 | 3.22 | 1.69 | 2.41 | 1.83 | 2.69 | 2.10 | 0.84 | 4.83 | 3.75 | 2.11 | 3.92 |
| | BLINK+SAT | 4.17 | 5.29 | 5.18 | 6.89 | 3.74 | 3.86 | 3.39 | 2.65 | 0.89 | 1.28 | 2.01 | 1.17 |
| | PCA | 8.66 | 5.06 | 5.01 | 2.94 | 8.28 | 4.60 | 4.62 | 3.54 | 6.74 | 4.80 | 4.72 | 3.50 |
| | Conv-PCA | 4.56 | 1.86 | 5.98 | 1.66 | 5.54 | 5.52 | 5.32 | 5.50 | 4.02 | 4.90 | 3.42 | 5.06 |
| | KS-PCA | 13.46 | 18.10 | 22.06 | 22.86 | 14.52 | 14.80 | 19.56 | 21.46 | 13.70 | 16.06 | 21.86 | 23.42 |
| | **KS-GNN** | **16.71** | **20.72** | **22.87** | **26.91** | **18.63** | **21.14** | **23.89** | **26.59** | **16.11** | **20.22** | **22.53** | **26.28** |
| Video | GraphSAGE | 0.21 | 0.15 | 0.00 | 0.15 | 0.03 | 0.06 | 0.12 | 0.00 | 0.21 | 1.20 | 0.78 | 0.84 |
| | BLINK+SAT | 4.34 | 3.78 | 4.56 | 5.22 | 3.39 | 2.71 | 3.67 | 3.28 | 2.29 | 1.98 | 1.55 | 1.20 |
| | PCA | 1.20 | 0.72 | 0.72 | 0.48 | 1.23 | 0.57 | 0.48 | 0.24 | 1.44 | 0.54 | 0.45 | 0.33 |
| | Conv-PCA | 5.85 | 4.47 | 3.96 | 4.50 | 4.89 | 6.90 | 6.15 | 9.66 | 6.21 | 7.50 | 6.84 | 8.52 |
| | KS-PCA | 8.60 | 8.29 | 11.66 | 10.60 | 6.53 | 6.25 | 9.99 | 12.43 | 5.94 | 7.02 | 7.04 | 9.37 |
| | **KS-GNN** | **9.27** | **9.05** | **12.53** | **17.67** | **10.23** | **8.15** | **12.75** | **20.92** | **8.27** | **8.57** | **16.77** | **19.58** |
| Toy | GraphSAGE | 0.74 | 0.88 | 5.11 | 4.21 | 4.24 | 12.49 | 6.50 | 6.10 | 1.45 | 4.39 | 6.71 | 0.09 |
| | BLINK+SAT | 9.46 | 9.58 | 9.57 | 9.66 | 7.208 | 11.5 | 9.30 | 12.02 | 7.44 | 8.96 | 10.09 | 6.48 |
| | PCA | 1.15 | 0.68 | 0.63 | 0.51 | 1.01 | 0.69 | 0.51 | 0.44 | 0.67 | 0.47 | 0.44 | 0.30 |
| | Conv-PCA | 21.34 | 21.99 | 23.76 | 25.4 | 19.17 | 19.72 | 20.21 | 25.22 | 16.99 | 21.61 | 23.95 | 24.90 |
| | KS-PCA | 27.23 | **27.73** | **31.58** | 33.79 | 25.78 | **28.94** | 31.04 | **32.82** | 18.35 | 22.03 | 25.50 | 26.25 |
| | **KS-GNN** | **28.56** | 26.85 | 30.55 | **34.28** | **26.65** | 27.76 | **32.27** | 32.25 | **21.78** | **27.41** | **25.55** | **30.17** |
| DBLP | GraphSAGE | 0.02 | 0.00 | 0.04 | 0.66 | 0.10 | 0.44 | 0.16 | 0.10 | 0.04 | 0.00 | 0.00 | 0.00 |
| | BLINK+SAT | 3.99 | 4.53 | 3.28 | 3.72 | 2.23 | 0.64 | 0.46 | 0.30 | 1.74 | 3.31 | 3.21 | 9.01 |
| | PCA | 2.38 | 1.90 | 1.76 | 1.46 | 2.26 | 2.00 | 1.54 | 1.36 | 2.24 | 1.62 | 1.40 | 1.16 |
| | Conv-PCA | 7.42 | 9.48 | 9.84 | 15.02 | 4.06 | 6.04 | 7.40 | 7.94 | 4.12 | 8.28 | 11.56 | 13.68 |
| | KS-PCA | 11.82 | 19.32 | 21.42 | 32.42 | 11.30 | 19.18 | 21.02 | 30.12 | 10.40 | 18.14 | 19.62 | 26.40 |
| | **KS-GNN** | **12.71** | **28.01** | **30.39** | **35.79** | **14.61** | **22.41** | **29.62** | **31.70** | **12.55** | **23.40** | **28.47** | **29.62** |

Table 4: Variances of KS-GNN results in graph with only missing keywords.

| Datasets | $r_w$ | 0.3 | | | | 0.5 | | | | 0.7 | | | |
|---|---|---|---|---|---|---|---|---|---|---|---|---|---|
| | $n_q$ | 3 | 5 | 7 | 9 | 3 | 5 | 7 | 9 | 3 | 5 | 7 | 9 |
| CiteSeer | **Hit@100** | 30.84 | 37.86 | 38.07 | 42.61 | 31.43 | 38.79 | 38.86 | 42.62 | 28.69 | 35.25 | 35.61 | 38.64 |
| | **Variance** | 0.43 | 0.53 | 0.37 | 0.27 | 0.27 | 0.45 | 0.81 | 0.96 | 0.23 | 0.55 | 0.41 | 0.25 |
| Video | **Hit@100** | 21.43 | 23.36 | 22.92 | 26.79 | 22.54 | 22.57 | 30.41 | 33.41 | 21.01 | 16.48 | 22.01 | 28.47 |
| | **Variance** | 0.09 | 0.21 | 0.12 | 0.41 | 0.12 | 0.31 | 0.16 | 0.46 | 0.11 | 0.24 | 0.24 | 0.44 |
| Toy | **Hit@100** | 28.56 | 29.85 | 29.55 | 34.28 | 24.65 | 29.16 | 31.27 | 33.25 | 21.78 | 27.41 | 25.55 | 30.17 |
| | **Variance** | 0.25 | 0.17 | 0.18 | 0.16 | 0.28 | 0.25 | 0.19 | 0.18 | 0.12 | 0.18 | 0.39 | 0.35 |
| DBLP | **Hit@100** | 16.21 | 24.94 | 29.55 | 33.51 | 16.52 | 22.73 | 26.85 | 30.69 | 15.57 | 24.15 | 27.12 | 29.06 |
| | **Variance** | 0.46 | 0.12 | 0.23 | 0.15 | 0.15 | 0.22 | 0.22 | 0.16 | 0.09 | 0.12 | 0.13 | 0.08 |

Table 5: Variances of KS-GNN results in graph with both missing keywords and edges ($r_e = 0.3$).

| Datasets | $r_w$ | 0.3 | | | | 0.5 | | | | 0.7 | | | |
|---|---|---|---|---|---|---|---|---|---|---|---|---|---|
| | $n_q$ | 3 | 5 | 7 | 9 | 3 | 5 | 7 | 9 | 3 | 5 | 7 | 9 |
| CiteSeer | **Hit@100** | 30.57 | 37.88 | 38.15 | 41.80 | 26.80 | 34.70 | 34.37 | 36.75 | 24.47 | 31.19 | 35.82 | 34.96 |
| | **Variance** | 0.22 | 0.47 | 0.47 | 0.66 | 0.62 | 0.91 | 0.79 | 0.83 | 0.29 | 0.51 | 0.43 | 0.42 |
| Video | **Hit@100** | 8.08 | 8.34 | 12.88 | 11.82 | 6.84 | 7.68 | 4.18 | 11.12 | 6.37 | 10.31 | 13.92 | 10.07 |
| | **Variance** | 0.05 | 0.15 | 0.32 | 0.12 | 0.02 | 0.24 | 0.17 | 0.12 | 0.04 | 0.24 | 0.19 | 0.12 |
| Toy | **Hit@100** | 13.82 | 13.28 | 14.38 | 15.22 | 12.51 | 12.54 | 12.68 | 12.59 | 9.41 | 8.99 | 12.83 | 11.15 |
| | **Variance** | 0.13 | 0.22 | 0.05 | 0.07 | 0.05 | 0.07 | 0.04 | 0.11 | 0.13 | 0.06 | 0.12 | 0.04 |
| DBLP | **Hit@100** | 15.91 | 20.49 | 25.09 | 29.04 | 15.79 | 19.86 | 22.15 | 29.71 | 12.07 | 17.18 | 19.23 | 24.19 |
| | **Variance** | 0.38 | 0.09 | 0.16 | 0.21 | 0.45 | 0.08 | 0.11 | 0.13 | 0.17 | 0.21 | 0.17 | 0.11 |

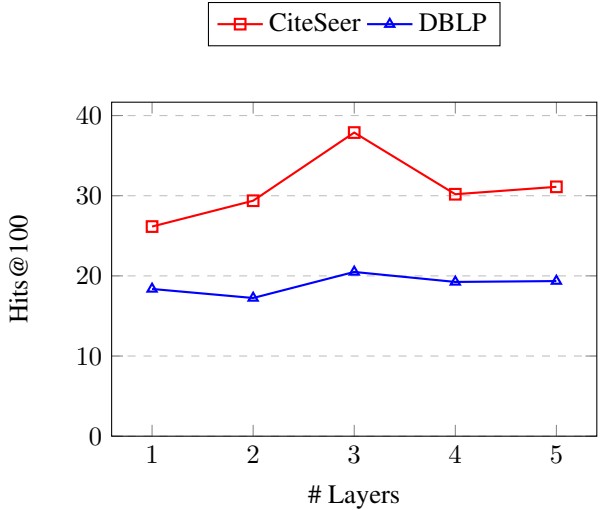

Figure 2: Sensitivity study of the number of layers for KS-GNN.

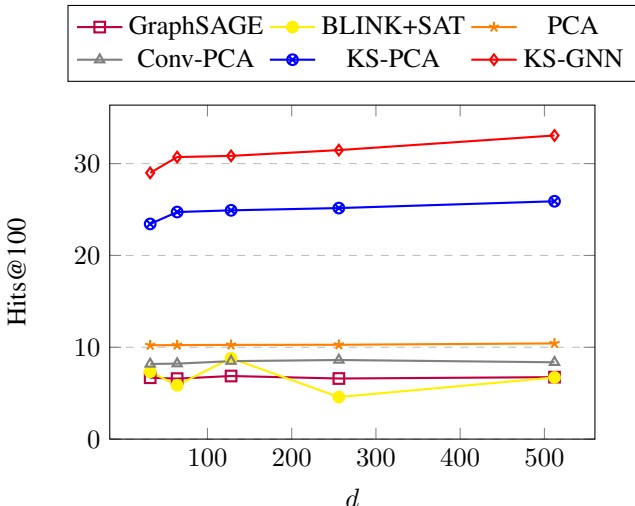

Figure 3: Sensitivity study of the output dimension of node embedding ($d$).

### E.5 Impact of $r_e$

To investigate the impact of $r_e$, We further conduct experiments on the CiteSeer dataset, where $r_e$ changes from $0$ to $0.5$ with a step of $0.1$, and other hyper-parameters are fixed according to the grid search algorithm results. The experiments results are shown in Fig. 4. From the figure, it can be observed that the performance of KS-GNN and KS-PCA decreases slightly with the increase of $r_e$. Other baseline methods with low performance are also not affected much by the increase of $r_e$. However, the performance of BLINK+SAT decreases significantly, as its link prediction model cannot accurately complete the missing edges.

### E.6 Analysis of Keyword Frequency Awareness

As discussed in Section 4.2, we propose a novel learning objective for training KS-GNN that aims to enhance its ability of keyword frequency awareness. Therefore, in the incomplete graph with $r_w = 0.3$ and $r_e = 0$, we conduct experiments which show the relation between the keyword frequency $c_i$ and the length of keyword embedding $||f(\mathbf{I}_i)||_2$. We compare the results by setting $\lambda_3$ to $1$ or $0$, which indicates whether to minimize $\mathcal{L}_3$ or not.

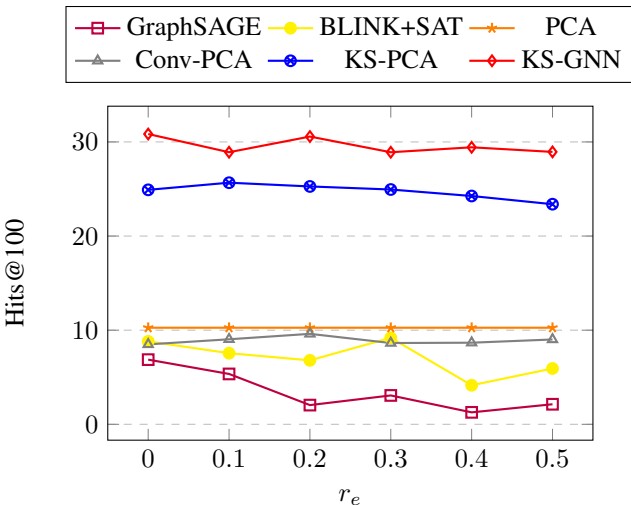

Figure 4: Impact of $r_e$ on CiteSeer.

As the figure shows, by minimizing $\mathcal{L}_3$, KS-GNN can significantly learn the keyword frequency awareness, which is reflected by the length of keyword embedding. It is presented that the keywords with high frequencies turn to be less important than before minimizing $\mathcal{L}_3$. Because compared with the long keyword embedding, shorter keyword embedding tends to be ignored during the query process. It is also interesting to notice that the lengths of some low-frequency keywords decrease. This is exactly what we expect since there are many low-frequency keywords in the graph, therefore it is meaningful to distinguish them according to their importance.

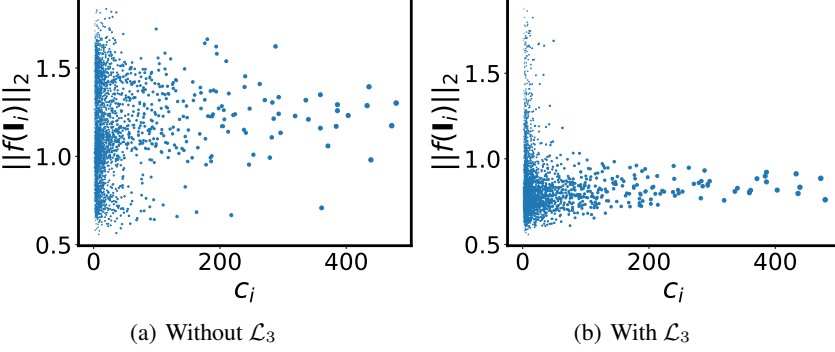

(a) Without $\mathcal{L}_3$          (b) With $\mathcal{L}_3$

Figure 5: Comparison of the ability of keyword frequency awareness by whether using $\mathcal{L}_3$ or not.

## F  Case Study

To show the ability of proposed KS-GNN to search a subgraph centered at the returned root node, we conduct a case study on the dataset Video with $r_e = 0.3$ and $r_w = 0.3$. We compare the subgraph retrieved by BLINK with that of KS-GNN. Specifically, when utilizing KS-GNN to search the subgraph, we can first find a root node $v_r$ which has the largest similar score to the query. Then, we use a BFS-based algorithm to check the neighbors of $v_r$. The details of this algorithm can be found in the Appendix. The results are shown in Fig. 6.

Given a query {Nintendo, Xbox, PC}, we aim to find a root node and a subgraph that covers the nodes containing the query keywords, and the best answer should have the minimum sum of the distances from the keywords to the root node. For instance, in Fig. 6 (a), the root node is B0001WN0MW and the subgraph is <B0001WN0MW, (B00005CFHJ, B00005MDZK)>. When we miss the edge between B00005CFHJ and B0001WN0MW as well as the keywords of B0001WN0MW as shown

**Query = {Nintendo, Xbox, PC}**

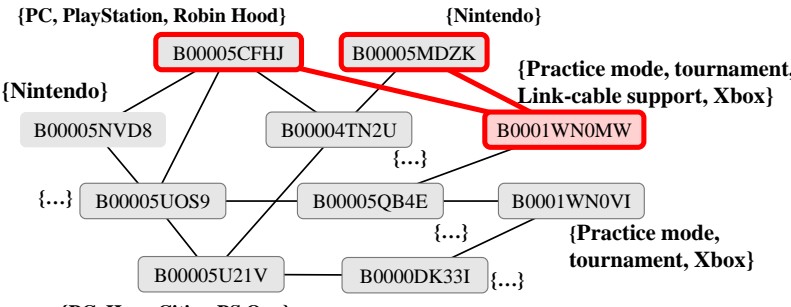

(a) The red circles and lines represent the best subgraph answer in the complete graph $G$.

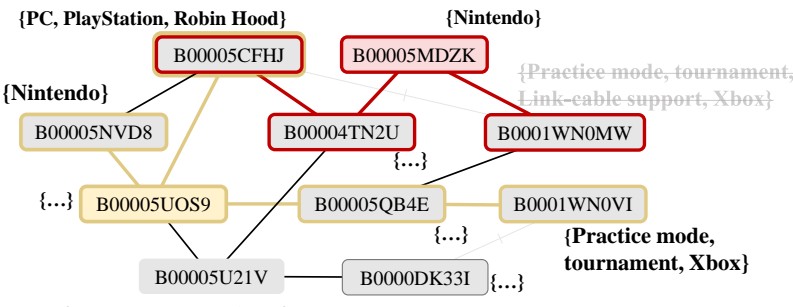

(b) The dark red circles and lines represent the subgraph retrieved by KS-GNN in the incomplete graph $G'$, while the yellow ones represent the subgraph retrieved by BLINK.

Figure 6: A case study which shows the answer subgraphs retrieved by BLINK and KS-GNN in $G'$.

in $G'$ of Fig. 6 (b), BLINK cannot retrieve the information on the node having missing keywords (i.e. B0001WN0MW), thus return a subgraph which is much different from the expected answer. However, our proposed KS-GNN is still able to retrieve the information on B0001WN0MW, and returns a similar subgraph which covers the expected answer as shown in Fig. 6 (b).