# OpenReview forum: "KS-GNN: Keywords Search over Incomplete Graphs via Graphs Neural Network"
_NeurIPS.cc/2021/Conference — NeurIPS 2021 Poster_

### Official Review · Reviewer_dWyG · 2021-07-16

**Rating:** 5
**Confidence:** 3

**Summary:**

This paper addressed an important problem that can answer queries in graph where some information is missing(edges and nodes). The proposed solution is based on auto-encoder and GNN. Here, the auto-encoder learns a low dimensional embedding for the node which was used to compute the score top-N ranked candidates based on similarity scores. Their method shows results better than other methods in the experiments.


**Ethical Concerns:**

Don't think there are ethical concerns over this work.

**Limitations And Societal Impact:**

Can not find a section on this on the paper.

**Main Review:**


Originality:

This paper creates a novel task of answering queries in graph where informations are missing. The work is a novel combination of known techniques.

Quality:

Scalability/Computational challenges are not discussed. What are we paying for to achieve this better result?

If we query for a complete graph what will be the performance for KS-GNN? In other words, is KS-GNN equally applicable when the graph is complete?

What type of missing information KS-GNN can handle is unclear? Random query generation process don't provide enough knowledge.

Ablation studies should be presented in the main paper. What is the effect of r_e and r_w to the performance? Are the numbers will be consistant if the r_e or r_w samples change?

Why hits@100 is presented at the main paper? Whereas it is more interesting to see the results for hits@10 or hits@50. If the query result is presented in top-100 is it a good result?

A small suggestion 4.1 Naive method is less important whereas more important parts i.e. ablation study (appendix E), algorithm (appendix D) will provide better insights to the end readers.

The learning objective from equation 8, it was not clear the impact of the three objectives on the learning process. The reader had to dig through appendix and found that, \mathcal{L}_3 is the key driving factor here. However, which parameter settings were presented at Table:4 at appendix? I was expecting the numbers should match for KS-GNN with the tables from the original paper Table 1 or Table 2.

How can we use KS-GNN when new keyword information is available? Do we need to retrain the complete model?

Clarity:

The paper was hard to follow. Even when the authors uses the figures, it is not referenced properly. For example - where the figure 4 was referenced in the paper? Also, the message from the figure was not understandable.

Significance:

The results are significant but the experimental setting and procedure has a lot of scope to improve on.




**Time Spent Reviewing:**

1

---

> ### Author Response · Authors · 2021-08-10
> **Response to Reviewer dWyG**
>
> We would like to thank the reviewers for their efforts, detailed review comments, great questions, and constructive suggestions. We will revise the manuscript to improve its clarity and reader friendliness. The following are our answers to specific questions:
>
> ---
>
> **Q1. Scalability/Computational challenges are not discussed. What are we paying for to achieve this better result?**
>
> A1: We discussed the query processing efficiency in Section E.1 in Appendix. We list the training time per epoch for four datasets as shown below. Specifically, the training time is mainly associated with the number of nodes (#Nodes) and the number of keywords (#Keywords), since the size of the input feature matrix and the number of parameters is related to them. In addition, we set the maximum number of training epochs to 100, which means the reported performance shown in Table 1 and Table 2 can be achieved by training our proposed KS-GNN within 2 minutes.
>
> | Dataset  | #Nodes | #Edges  | #Keywords | Training Time per Epoch (s) |
> | :-------- | ------: | -------: | ---------: | :---------------------------: |
> | DBLP     | 32,361 | 69,448  | 4,094     | 0.29                        |
> | Toy      | 20,682 | 224,603 | 4,114     | 0.45                        |
> | Video    | 20,882 | 52,886  | 11,514    | 0.69                        |
> | Citeseer | 3,327  | 9,104   | 3,703     | 0.06                        |
>
> ---
>
> **Q2. If we query for a complete graph what will be the performance for KS-GNN? In other words, is KS-GNN equally applicable when the graph is complete?**
>
> A2: In real-world applications, the graphs are usually incomplete due to privacy issues or the difficulties of collecting the data. To perform the keyword search task on complete graphs, our proposed naïve method Conv-OH can return the same exact answer as the conventional methods such as BLINKS. As there is dimensionality reduction in KS-GNN, the original keyword information could not be fully reserved. Therefore, KS-GNN might not find the same answer as BLINKS if KS-GNN outputs low-dimensional node embeddings.
>
> In addition, our work has a different research aim from the existing graph keyword search methods — we focus on incomplete graphs and the main goal is to improve the accuracy, while existing methods like BLINKs implicitly assume the graph is complete and focus on improving the query efficiency.
>
>
> ---
>
> **Q3. What type of missing information KS-GNN can handle is unclear? Random query generation process don't provide enough knowledge.**
>
> A3: According to the problem definition, our KS-GNN could handle two kinds of missing information —  missing keywords and missing edges. Following the experimental settings of [1,2], we employ the same randomly sampling method to hide the information on the original datasets to simulate the missing information scenarios in the real world. We use the random query generation to avoid the potential bias, and we aim to show that our model can always achieve much better results on graphs with missing keywords and missing edges.
>
> We also have conducted additional experiments using the ten queries provided by BLINKS [3] over the DBLP dataset, and we observe similar experimental results which show that our KS-GNN model outperforms the baseline methods consistently. We will include the full results in the Appendix of the final version.
>
> [1] You, J., Ma, X., Ding, D.Y., Kochenderfer, M. and Leskovec, J., 2020. Handling Missing Data with Graph Representation Learning. Neural Information Processing Systems (NeurIPS).
>
> [2] Chen, X., Chen, S., Yao, J., Zheng, H., Zhang, Y. and Tsang, I.W., 2020. Learning on attribute-missing graphs. IEEE Transactions on Pattern Analysis and Machine Intelligence.
>
> [3] He, H., Wang, H., Yang, J., & Yu, P. S., 2007. Blinks: ranked keyword searches on graphs. In Proceedings of the ACM SIGMOD international conference on Management of data
>
> ---
>
> **Q4. Ablation studies should be presented in the main paper. What is the effect of r_e and r_w to the performance? Are the numbers will be consistant if the r_e or r_w samples change?; A small suggestion 4.1 Naive method is less important whereas more important parts i.e. ablation study (appendix E), algorithm (appendix D) will provide better insights to the end readers; The learning objective from equation 8, it was not clear the impact of the three objectives on the learning process. The reader had to dig through appendix and found that, \mathcal{L}_3 is the key driving factor here.**
>
> A4: Thanks for your suggestion! We will present the results of the ablation study in the revised final version. In Table 1, $r_e$ is set to 0 and $r_w$ ranges from 0.3 to 0.7. In Table 2, $r_e$ is set to 0.3 and $r_w$ ranges from 0.3 to 0.7. The numbers are consistent if the $r_e$ or $r_w$ samples change. We have conducted additional experiments to study the variance of the results of KS-GNN by varying the seed of edge and keyword sampling (10 runs for each set of experiments). It can be observed that the variance values are always small.
>
> ---
>
> **Q5. Why hits@100 is presented at the main paper? Whereas it is more interesting to see the results for hits@10 or hits@50. If the query result is presented in top-100 is it a good result?**
>
> A5: Because of the page limitation, we put the results for Hits@10 and Hits@50 in the Appendix. In fact, Hits@10 can show better results in terms of the improvement over baseline methods. We computed the average gain, i.e. $(Hits_{KSGNN} - Hits_{baseline})/Hits_{baseline}$, where $Hits_{KSGNN}$ denotes the performance of our proposed KS-GNN and $Hits_{baseline}$ denotes the best performance of the baseline methods, over three tables. As shown in the below table, the gain rates of Hits@10 are always larger than those of Hits@50 and Hits@100, while the performance of Hits@50 is better than Hits@100 mostly.
>
> |         | Citeseer | Video | Toy  | DBLP |
> | :------- | --------: | -----: | ----: | ----: |
> | Hits@10  | $\mathbf{1.03}$     | $\mathbf{3.36}$  | $\mathbf{1.09}$ | $\mathbf{1.91}$ |
> | Hits@50  | 0.43     | 0.46  | 0.32 | 1.10 |
> | Hits@100 | 0.36     | 0.66  | 0.24 | 0.69 |
>
> ---
>
> **Q6. However, which parameter settings were presented at Table:4 at appendix? I was expecting the numbers should match for KS-GNN with the tables from the original paper Table 1 or Table 2.**
>
> A6: In terms of the parameter setting in Table 4, we set $r_w$=0.3, $r_e$=0.3 and $n_q$=5 where it is mentioned in lines 102-103 in Appendix. The numbers are indeed matched with Table 2 in the original paper.
>
> ---
>
> **Q7. How can we use KS-GNN when new keyword information is available? Do we need to retrain the complete model?**
>
> A7: If the new keyword information exists only in the training set or queries, we do not need to retrain the model. Otherwise, we need to retrain KS-GNN, because the input feature matrix $\mathbf{X}$ is the one-hot encoding of keywords, and the appearance of new keywords will change the size of the input feature matrix correspondingly. This is the problem that other GNN-based models with the input feature matrix as input also have to face.
>
> ---
>
> **Q8: The paper was hard to follow. Even when the authors uses the figures, it is not referenced properly. For example - where the figure 4 was referenced in the paper? Also, the message from the figure was not understandable.**
>
> A8: Thanks for the reviewer’s suggestion. The figure should be referenced in line 350. We will update the figure in the revised final version. As for Figure 4, we discuss the motivation and learning objective in lines 194-204 and lines 261-270, respectively. By minimizing Eq. (7), we aim to differentiate the lengths of keyword embeddings according to their keyword frequencies, and the result of this is visualized in Figure 4.
>
> ---
>
> Thank you very much for your thorough review and for providing useful suggestions in terms of citations. In the light of these clarifications, we would appreciate it if the reviewer confirmed that all the concerns had been addressed and, if so, reconsider the assessment.

---

### Official Review · Reviewer_kP3a · 2021-07-16

**Rating:** 6
**Confidence:** 3

**Summary:**

The paper proposes a method to run keyword searches over graphs that are incomplete. The proposal is that of encoding the graph into a latent space that allows the reconstruction of the graph and then the subsequent application of graph search mechanisms.

**Limitations And Societal Impact:**

Authors should run an analysis to put confidence intervals around results in tables 1 and 2.

**Main Review:**

Strengths:
* The paper is well written
* Experiments show that the method can effectively solve the problem of searching in graphs when the information is missing about the connection between nodes.

Weaknesses:
* The motivations are not clearly stated. Keyword search on graphs is usually required for solving XML queries data. Can you better motivate the problem by listing some important applications of keyword search on incomplete graphs?
* The description of the problem is not clear. In the beginning, you propose to first encode the graph and then use the latent representation to overcome the problem of missing information (Figure 2). Then, in Figure 3, you add some other layers, and the way these components are integrated is written in Eq. (8) which is at the very end of the description and it is confusing.
* The complexity of the query processing phase is very high (dN) and it might be impractical in a real-world scenario
* Given the stochastic nature of the missing info problem, I would have appreciated an analysis of the variance of the results varying the seed of edge and keyword sampling.

Comments on the content
* It is not clear in the problem definition (Line 110) why you introduce r_w and r_e given that they are not part of the definition


**Time Spent Reviewing:**

6

---

> ### Author Response · Authors · 2021-08-10
> **Response to Reviewer kP3a**
>
> We would like to thank the reviewers for their efforts, positive review comments, and suggestions. We will revise the manuscript to improve its clarity and reader friendliness. The following are our answers to specific questions:
>
> ---
>
> **Q1. The motivations are not clearly stated. Keyword search on graphs is usually required for solving XML queries data. Can you better motivate the problem by listing some important applications of keyword search on incomplete graphs?**
>
> A1: Besides solving XML queries data, keyword search is also widely used on many other types of graphs, such as knowledge graphs [2] and general attributed graphs [3, 4]. Keyword search on graphs has been applied in various important applications such as POIs recommendation and keyword-aware routing as pointed out by a survey [1]. However, in real-world applications, the graphs are usually incomplete (including XML [5], knowledge graphs [6], and attributed graphs [7]).
>
> Therefore, we propose KS-GNN to solve the keyword search problem on incomplete graphs. To our best knowledge, this is the first work on keyword search in graphs with missing information. All the existing conventional methods (such as [8, 9]) cannot solve the problem on incomplete graphs. (We will include more citations to discuss this in the final version.)
>
> [1] Yang, J., Yao, W., & Zhang, W. (2021). Keyword Search on Large Graphs: A Survey. Data Science and Engineering, 6(2), 142-162.
>
> [2] Shi, Y., Cheng, G., & Kharlamov, E. (2020, April). Keyword search over knowledge graphs via static and dynamic hub labelings. In Proceedings of The Web Conference 2020 (pp. 235-245).
>
> [3] Ghanbarpour, A., & Naderi, H. (2018). An attribute-specific ranking method based on language models for keyword search over graphs. IEEE Transactions on Knowledge and Data Engineering, 32(1), 12-25.
>
> [4] Bryson, S., Davoudi, H., Golab, L., Kargar, M., Lytvyn, Y., Mierzejewski, P., ... & Zihayat, M. (2020). Robust keyword search in large attributed graphs. Information Retrieval Journal, 23(5), 502-524.
>
> [5] Barceló, P., Libkin, L., Poggi, A., & Sirangelo, C. (2010). XML with incomplete information. Journal of the ACM (JACM), 58(1), 1-62.
>
> [6] Wang, Q., Mao, Z., Wang, B., & Guo, L. (2017). Knowledge graph embedding: A survey of approaches and applications. IEEE Transactions on Knowledge and Data Engineering, 29(12), 2724-2743.
>
> [7] Chen, X., Chen, S., Yao, J., Zheng, H., Zhang, Y., & Tsang, I. W. (2020). Learning on attribute-missing graphs. IEEE Transactions on Pattern Analysis and Machine Intelligence.
>
> [8] He, H., Wang, H., Yang, J., & Yu, P. S. (2007, June). Blinks: ranked keyword searches on graphs. In Proceedings of the 2007 ACM SIGMOD international conference on Management of data (pp. 305-316).
>
> [9] Jiang, M., Fu, A. W. C., & Wong, R. C. W. (2015, May). Exact top-k nearest keyword search in large networks. In Proceedings of the 2015 ACM SIGMOD international conference on management of data (pp. 393-404).
>
> ---
> **Q2. The description of the problem is not clear. In the beginning, you propose to first encode the graph and then use the latent representation to overcome the problem of missing information (Figure 2). Then, in Figure 3, you add some other layers, and the way these components are integrated is written in Eq. (8) which is at the very end of the description and it is confusing.**
>
> A2: We introduce the problem in Section 3 and provide the detailed problem definition in lines 110 – 114. As stated in line 204, Fig. 3 illustrates the message passing and aggregation framework of our proposed KS-GNN model, and this figure matches the Eq.(5) instead of Eq.(8). We will refer to Fig.3 around Eq. (5) for a better presentation of our model. Eq. (8) describes the loss function and is proposed to minimize Eq.(4), Eq.(6) and Eq.(7) simultaneously.
>
> It is worth noting that the naïve method Conv-OH illustrated in Fig. 2 employs traditional message passing and aggregation mechanisms of the general graph neural networks (GNN). The reason for presenting Conv-OH first is to explain why our proposed KS-GNN employing a different message passing and aggregation framework can avoid the high space cost of Conv-OH while achieving good performance.
>
> ---
>
> **Q3. The complexity of the query processing phase is very high (dN) and it might be impractical in a real-world scenario**
>
> A3: We process the query with stored N $d$-dimensional node embeddings and thus the size is $N \times d$. Here, $O(dN)$ is the space complexity, and it is the minimum space cost to keep the embeddings of all the nodes. Note that all existing node embedding based methods have this space cost for storing the embedding vectors. For example, given $d$=64 and assuming that we use 4 bytes to store the value for each dimension, for a graph with even one billion nodes, 256G is required by KS-GNN, which can be easily fit into nowadays high-performance computing servers.
>
> ---
>
> **Q4. Given the stochastic nature of the missing info problem, I would have appreciated an analysis of the variance of the results varying the seed of edge and keyword sampling.**
>
> A4: We include the variance of the results of KS-GNN varying the seed of edge and keyword sampling as below:
>
> | Dataset  | $r_e$     | 0     |       |       | 0.3   |       |       |
> | -------- | -------- | -----: | -----:  | -----:  | -----:  | -----:  | -----:  |
> |          | $r_w$    | 0.3   | 0.5   | 0.7   | 0.3   | 0.5   | 0.7   |
> | Citeseer | Hit@100   | 31.21 | 33.19 | 32.56 | 29.97 | 30.17 | 30.61 |
> |          | Variance | 0.07  | 0.32  | 0.28  | 0.21  | 0.45  | 0.31  |
> | Video    | Hit@100 | 21.43 | 22.54 | 21.01 | 12.81 | 10.01 | 10.34 |
> |          | Variance | 0.04  | 0.05  | 0.05  | 0.01  | 0.02  | 0.02  |
> | Toy      | Hit@100 | 25.42 | 29.26 | 18.82 | 12.83 | 12.09 | 8.27  |
> |          | Variance | 0.20  | 0.11  | 0.09  | 0.01  | 0.01  | 0.03  |
> | DBLP     | Hit@100 | 24.41 | 22.21 | 20.44 | 22.07 | 17.26 | 17.12 |
> |          | Variance | 0.03  | 0.02  | 0.01  | 0.11  | 0.10  | 0.03  |
>
> The table shows that the variance of experimental results over 10 runs on four datasets with $n_q$=3 (number of query keywords),
> $r_e$ being set to 0 and 0.3, and $r_w$ ranging from 0.3 to 0.7, respectively. From this table, we can see that the variances are always very small. We will provide the variances of full Table 1 and Table 2 in the revised Appendix.
>
> ---
>
>
> **Q5. It is not clear in the problem definition (Line 110) why you introduce r_w and r_e given that they are not part of the definition**
>
> A5: As shown in lines 111 – 112, $r_w$ and $r_e$ indicates the proportion of nodes with missing keywords and the proportion of missing edges in G, respectively. Our problem is defined over incomplete graphs. Therefore, they are necessary to be part of the definition for defining the incomplete graph G’, and we introduce these two proportions in line 110.
>
> ---
>
> Thank you very much for your thorough review and for providing useful suggestions in terms of citations. We hope that our explanations have successfully cleared your concerns.

---

> > ### Comment · Reviewer_kP3a · 2021-08-18
> > **I have read your answer**
> >
> > Thanks for replying to my questions. I believe many of my questions were due to the current status of the paper. I would encourage authors to change the text of the article in order to make it more understandable by readers. I believe the paper has nice results. but I will stick to my current score, as another concern of mine could be the impact of the paper on the general audience.

---

### Official Review · Reviewer_ZPE7 · 2021-07-20

**Rating:** 7
**Confidence:** 3

**Summary:**

This paper introduces KS-GNN, Keywords Search GNN over Incomplete Graphs. More specifically, KS-GNN employs joint training on autoencoder (dimension deduction), keyword-based node similarity in subgraphs, and keyword frequency regularization to facilitate a complex keyword search task on incomplete graphs (missing keywords in node attributes and edges). Experimental results have proven the effectiveness of the proposed KS-GNN over multiple baseline approaches compared to various standard benchmarks.

**Ethical Concerns:**

Not applicable.

**Limitations And Societal Impact:**

Not applicable.

**Main Review:**

Strengths:
1. Well-motivated challenging task of keyword search over incomplete graphs.
2. KS-GNN is technically sound and capable of modeling incompleteness, compared to Conv-OH and Conv-PCA.
3. Extensive experiments on a large number of settings are provided.

Weaknesses:
1.The problem formulation is somewhat unclear in the statement and introduction examples.
2. More baselines or self variants should be compared to better prove the effectiveness.

Detailed comments:
1. The problem definition of keyword search on incomplete graphs is ambiguous and confusing. The KS-GNN mostly optimizes on node similarity and the inference stage tends to select the top-k most similar results towards the query keyword set. However, the problem itself seems more like a combinatorial one, or say, set optimization, node-set selection with minimal distance measurement. Are the targets here equivalent?
2. The baseline approach seems much inferior to KS-GNN. It would be great to include some variants of the KS-GNN that delete some of the module or training objectives to confirm the contribution of each component.
3. Table 2 with missing edges is supposed to be more challenging than the task in Table 1. However, a lot of models perform even better (or comparable) which seems strange. Also,  the claim of “KS-GNN has no significant effect” does not apply to the Toy and Video datasets for correctness.
4. It is hard to conclude from Figure 4 on the benefit of keyword frequency regularization. That is, it’s better to show the performance scores along with the visualization.
5. Figure 3: The notations of the figure (especially function f and g) are confusing.

**Time Spent Reviewing:**

1.5

---

> ### Author Response · Authors · 2021-08-10
> **Response to Reviewer ZPE7**
>
> We would like to thank the reviewers for their efforts, positive review comments, and suggestions. We will revise the manuscript to improve its clarity and reader friendliness. The following are our answers to specific questions:
>
> ---
>
> **Q1. The problem definition of keyword search on incomplete graphs is ambiguous and confusing. The KS-GNN mostly optimizes on node similarity and the inference stage tends to select the top-k most similar results towards the query keyword set. However, the problem itself seems more like a combinatorial one, or say, set optimization, node-set selection with minimal distance measurement. Are the targets here equivalent?**
>
> A1: Yes, the targets here are equivalent. The keyword search on graphs usually retrieves a set of nodes, but as shown in BLINKS (one of the most well-known graph keyword search methods), it can be converted to retrieving a single node (the details and an example are introduced in lines 104-109). As introduced in the problem definition (lines 110-114), we employ the same scoring function as that of BLINKS, and our problem and BLINKS have the same searching objective. Our proposed naïve method Conv-OH can return the same answer as BLINKS, and based on this we further propose the more powerful KS-GNN.
>
> ---
>
> **Q2. The baseline approach seems much inferior to KS-GNN. It would be great to include some variants of the KS-GNN that delete some of the module or training objectives to confirm the contribution of each component.**
>
> A2: Thank you for the suggestion. We have conducted experiments of ablation study to investigate how each component in Eq. (8) affects our proposed KS-GNN’s performance. The ablation study is presented in Section E.6 in the Appendix of the paper. Please find the experimental results in Table 4 in the Appendix.
>
> ---
>
> **Q3. Table 2 with missing edges is supposed to be more challenging than the task in Table 1. However, a lot of models perform even better (or comparable) which seems strange. Also, the claim of “KS-GNN has no significant effect” does not apply to the Toy and Video datasets for correctness.**
>
> A3: Thank you for pointing out this inaccurate claim, and we will improve this in the revised final version. As shown in Tables 1 and 2, this problem is often encountered when running the baselines GraphSAGE and BLINK+SAT on the dataset CiteSeer.
>
> The first reason may be that both GraphSAGE and SAT are GNN-based models and gather information from neighbours without considering the desiderata proposed in our paper (lines 180-204). Therefore, these two baselines may gather noisy information from neighbours and are not powerful enough to complete the proposed task, and the effect of missing edges cannot be reflected based on their very poor performance.
>
> The second reason could be due to that there are much more keywords on each node on CiteSeer than on the other datasets. We have computed the average number of keywords on each node for four datasets, and we found that the value on CiteSeer is around 31.6, which is much larger than the values on the other datasets (around 5). Hence, when querying less than 10 keywords, the missing edge information may improve the performance of GraphSage and BLINK+SAT because of reducing the noises from neighbours and focusing more on the target node itself. Therefore, we can observe these “strange” results more often on CiteSeer.
>
> In the original claim, we tended to compare our proposed KS-GNN with the competitive baseline methods such as Conv-PCA and Conv-rPCA. In the revised final version, we will improve our descriptions and provide more analysis about this issue.
>
> ---
>
> **Q4. It is hard to conclude from Figure 4 on the benefit of keyword frequency regularization. That is, it’s better to show the performance scores along with the visualization.**
>
> A4: Table 4 in the Appendix shows the performance improvement brought by keyword frequency regularization (Eq. (7)), and we will move this ablation study section in the main paper for better representation in the final version.
>
> ---
>
> **Q5. Figure 3: The notations of the figure (especially function f and g) are confusing.**
>
> A5: The encoder (denoted as $f$) used in KS-GNN takes the one-hot encoding matrix $\mathbf{X}$ as input and outputs node embeddings (denoted as $\mathbf{h}$). The decoder (denoted as g) takes $\mathbf{h}$ as input and aims to reconstruct $\mathbf{X}$. In KS-GNN, the messages passed between nodes are the node embeddings. As shown in Fig. 3, we first use the decoder $g$ to reconstruct the one-hot encodings (representing the keywords received from the neighbors) from the received messages and then perform the aggregation over the reconstructed vectors, and next, we use the encoder $f$ to compute the node embeddings for being passed in the next layer. We can add more illustrations of this figure in the final version.
>
> ---
>
> Thanks again for your supportive comments. We hope that our explanations have successfully cleared your concerns.

---

> > ### Comment · Reviewer_ZPE7 · 2021-08-20
> > **Author's comments have been read.**
> >
> > I have read the author's replies for my review concerns as well as others. The authors generally addressed my questions and provided a satisfactory response. I would stick to my current score evaluation considering the innovation, significance and clarity.

---

### Decision · Program_Chairs · 2021-09-27

**Decision:**

Accept (Poster)

**Comment:**

This paper addresses the novel and technically interesting problem of answering queries on graphs with missing information such as edges and nodes.
The proposed problem is well-motivated and the proposed method based on autoencoders and GNNs, although being a combination of known techniques, sounds reasonable, and is considered as a solid technical contribution, and also  the experimental evaluations are fairly thorough in a variety of settings.
Overall, this paper is expected to be of interest to the NeurIPS community, and worth being accepted.